# Surface Renewal as a Significant Mechanism for Dust Emission

Jie Zhang[1,2], Zhenjiao Teng[2], Ning Huang[1,2], Lei Guo[2], Yaping Shao[3]

[1]Key Laboratory of Mechanics on Disaster and Environment in Western China (Lanzhou University), Ministry of Education, 730000 Lanzhou, China
[2]School of Civil Engineering and Mechanics, Lanzhou University, 730000 Lanzhou, China
[3]Institute for Geophysics and Meteorology, University of Cologne, 50937 Cologne, Germany

*Correspondence to*: Ning Huang (huangn@lzu.edu.cn)

**Abstract.** Wind-tunnel experiments of dust emissions from different soil surfaces are carried out to better understand dust
emission mechanisms. The effects of surface renewal on aerodynamic entrainment and saltation bombardment are analysed
in detail. It is found that flow conditions, surface particle motions (saltation and creep), soil dust content and ground
obstacles all strongly affect dust emission, causing its rate to vary over orders of magnitude. Aerodynamic entrainment is
highly effective, if dust supply is unlimited, as in the first 2-3 minutes of our wind-tunnel runs. While aerodynamic
entrainment is suppressed by dust supply limit, surface renewal through the motion of surface particles appears to be an
effective pathway to remove the supply limit. Surface renewal is also found to be important to the efficiency of saltation
bombardment. We demonstrate that surface renewal is a significant mechanism affecting dust emission and recommend that
this mechanism be included in future dust models.

**Keywords:** dust emission; surface renewal; aerodynamic entrainment; wind tunnel; supply limit

## 1. Introduction

Three dust emission mechanisms have been identified, including (1) aerodynamic entrainment; (2) saltation bombardment;
and (3) aggregates disintegration (Shao, 2008; Kok et al., 2012; Újvári et al., 2016). In spite of much research effort, many
questions remain unanswered in relation to the process of dust emission. For example, in most existing dust emission
schemes, aerodynamic entrainment is assumed to be small and negligible. It is however questionable, to what extent and
under what conditions this assumption is justified, because there are hardly any data which enable a rigorous comparison of
aerodynamic entrainment from natural soil surfaces with the other dust emission mechanisms. For natural soils, dust
emission is usually "supply limited" (Shao, 2008; Macpherson et al., 2008; Újvári et al., 2016), i.e., the emission is limited
by the availability of free particles on the soil surface, rather than by the shear stress that wind exerts (note that 'supply
limited' in this paper only refers to a lack of supply of fine soil particles, but not saltators). However, "supply limit" is not a
quantified term in published emission models, as little is known about its spatial and temporal variations. The argument for
the neglect of aerodynamic entrainment is that dust particles have relatively large cohesive forces and are resistant to
aerodynamic lift, and thus saltation bombardment and aggregates disintegration are the dominant mechanisms for dust
emission (Greeley and Iversen, 1985; Shao et al., 1993). Researchers have noted there are obvious differences in dust

emission from disturbed and undisturbed soils (Macpherson et al., 2008, MP2008 hereafter). This is because soil disturbance replenishes dust supply to aerodynamic entrainment and modifies the aerodynamic properties of the surface, which may enhances momentum transfer from the atmosphere to the surface. Further, in existing dust models, the conditions of the surface subject to erosion are assumed to be stationary. In reality, during an erosion event, surface self-disturbance occurs due to top soil removal and particle impact, i.e., a surface renewal process takes place, which in general enhances the supply of dust for aerodynamic entrainment. We argue that under the conditions of strong surface renewal, aerodynamic entrainment may be a significant mechanism for dust emission.

In this work, we simulate three typical landforms in a wind-tunnel experiment, namely, a farmland surface, a desert surface and a loess surface (see Section 3 for details). We then seek to quantify the contributions of three dust emission mechanisms to the total dust flux for the different landforms. Using the wind-tunnel observations, we demonstrated that supply limit of free dust is a major factor which suppresses aerodynamic entrainment, but surface renewal through saltation and creep provides an important pathway to enhance the free dust supply for aerodynamic entrainment. Thus, for surfaces with strong renewal and sufficient free dust supply, aerodynamic entrainment becomes a non-negligible process for dust emission.

## 2. Background of Dust Emission Mechanisms

In general, dust emission flux, $F$, is considered to be caused by three mechanisms and can be expressed as

$$F = F_a + F_b + F_c \tag{1}$$

where $F_a$, $F_b$, $F_c$ are respectively the fluxes arising from aerodynamic entrainment, saltation bombardment and aggregates disintegration. $F_a$ is directly related to surface shear stress, while $F_b$ and $F_c$ depend on saltation. Here we will briefly review the studies on dust emission mechanisms and summarize the dust emission flux formulations. We will then introduce the basic assumptions of our study.

### 2.1 Aerodynamic Entrainment

Aerodynamic entrainment refers to direct dust uplift from the surface into the atmosphere by aerodynamic forces. It has been suggested that the dust flux arising from aerodynamic entrainment is insignificant, because aerodynamic lift force for small particles is in general small compared to inter-particle adhesion. Loosmore and Hunt (2000, LS2000 hereafter) suggested based on their wind-tunnel experiments that

$$F_a = 3.6 \, u_*^3 \tag{2}$$

where $F_a$ is in $\mu g \cdot m^{-2} \cdot s^{-1}$. Shao (2008) suggested that, inter-particle cohesive force is a stochastic variable, such that there always exists in nature a proportion of dust which is free, i.e., dust for which inter-particle cohesion is weak. Several studies have demonstrated that $F_a$ is not always negligible (Kjelgaardet al., 2004; MP2008; Klose and Shao, 2012; Sweeney and

Mason, 2013), but the key factors which determine aerodynamic entrainment remain poorly understood. Moreover, Loosmore and Hunt (2000) conducted the wind tunnel experiments by using "Arizona Test Dust" (ISO-12103-1) to produce very smooth test beds. The investigation of dust emission caused by aerodynamic entrainment over natural and rough surfaces is still lacking.

## 2.2 Saltation Bombardment

Saltation bombardment is considered as the central mechanism of dust emission and has been extensively studied. Based on field experiments (Gillette, 1974, 1977 and 1981), Gillette & Passi (1988, GP88 hereafter) proposed an empirical formula for dust flux due to saltation bombardment, $F_b$, as a function of friction velocity

$$F_b = c \cdot u_*^n (1 - \frac{u_{*t}}{u_*}) \tag{3}$$

where $c$ is an empirical constant and $n$ is suggested to be 4 (GP88). According to existing field measurements, Shao (2008) stated that dust emission flux can be proportional to $u_*^n$ but with $n$ varying between 2.9 and 4.4 and depending on soil type and soil-surface conditions. Many other studies have been carried out on sandblasting dust emission. For example, Marticorena and Bergametti (1995) suggested that dust emission flux is dependent on streamwise saltation flux and soil clay content, and Alfaro and Gomes (2001) suggested that sandblasting results in dust emission from three separate lognormal particle-size modes, and the contribution of the modes depends on the particle binding energy and the kinetic energy of impacting saltators.

Based on the wind-tunnel observations by Rice et al. (1996a, b) and Shao (1993, 1996), Lu and Shao (1999, LS99 hereafter) and Shao (2000, 2001) argued that a blasting saltator, upon its impact, causes a bombardment effect which results in dust emission. The latter authors derived a physical expression for dust emission by saltation

$$F_b = \frac{c_b g \xi \rho_b}{P} \left(1 + 14 u_* \sqrt{\frac{\rho_b}{P}}\right) Q \tag{4}$$

where $c_b$ is a constant, $g$ is gravitational acceleration, $\xi$ is the mass fraction of dust inside the crater, $\rho_b$ is the soil bulk density, $P$ is the horizontal component of soil plastic pressure determined by soil property and $Q$ represents saltation intensity which can be estimated by using the Owen model as shown in the next section.

## 2.3 Aggregates Disintegration

Studies on aggregates disintegration are rare. Shao (2001) presented a dust emission model which accounts for both the effect of saltation bombardment and aggregates disintegration. This model, as simplified in Shao (2004, S04 hereafter), can be summarized as follows:

$$F_b + F_c = \sum_{i=1}^{I} F(d_i) \tag{5}$$

$$F(d_i) = \int_{d_1}^{d_2} F(d_i, d_s) p(d_s) \, \delta d_s \tag{6}$$

$$F(d_i, d_s) = c_y \xi_{fi} \big[ (1 - \gamma) + \gamma \sigma_p \big] (1 + \sigma_m) \frac{g Q(d_s)}{u_*^2} \tag{7}$$

$$Q(d_s) = c_0 \frac{\rho}{g} u_*^3 \left( 1 - \frac{u_{*t}^2(d_s)}{u_*^2} \right), \text{ with } c_0 = 0.25 + \frac{v_t}{3 u_*} \qquad \text{(Owen, 1964)} \tag{8}$$

where $d_i$ is the particle size of the $i$th bin out of the total $I$ bins, $d_s$ is the particle size of the saltator, $F(d_i)$ represents the flux of dust of size $d_i$, and $F(d_i, d_s)$ represents the fraction of $F(d_i)$ which is caused by saltators of size $d_s$. $d_1$ and $d_2$ are the lower and upper limits of $d_s$. $p(d_s) = \gamma p_m(d_s) + (1 - \gamma) p_f(d_s)$ is the particle size distribution of $d_s$, $p_m(d_s)$ and $p_f(d_s)$ are respectively the distributions of saltators with statuses of minimally and fully disturbances, $\gamma = \exp[-(u_* - u_{*t})^3]$ (Shao et al., 2011), $c_y$ is a dimensionless coefficient, $\xi_{fi}$ is total dust fraction of the $i$th bin, $\sigma_p$ is the ratio of aggregated dust to free dust, $\sigma_m$ is the mass ratio of ejectiles to saltators (i.e., bombardment efficiency) derived from the saltation model by Lu and Shao (1999). Saltation intensity $Q(d_s)$ is evaluated by Owen model (Equation 8, where $\rho$ is air density, 1.25 kg·m$^{-3}$) and the particle terminal velocity is calculated by $v_t = 1.66(\sigma_\varphi g d_s)^{1/2}$ (Shao, 2008), with particle-to-air density ratios $\sigma_\varphi$=2120. Equation (5) sums the dust fluxes of all size bins and Equation (6) gives the dust flux of particles in the $i$th bin. In the end, emission dust flux is found to be proportional to $Q(d_s)$, but the proportionality depends on soil texture and soil plastic pressure. Further simplification indicates that at high soil plastic pressure (>3×10$^5$ Pa), $\sigma_m$ becomes negligibly small (<0.1) under normal wind conditions, and saltation bombardment diminishes to such an extent that aggregates disintegration prevails.

Kok et al. (2014) proposed a dust emission parameterization by using a combination of theory and numerical simulations. Their model primarily considers dust emission by aggregates disintegration and is in good agreement with a quality-controlled compilation of experimental measurements. But it is difficult to distinguish the contributions of the different dust emission mechanisms from experimental data (especially for field measurement). And it appears to be untenable to assume that dust emission is mainly caused by sandblasted fragmentation. We argue that aerodynamic entrainment should not be simply ignored and a series wind tunnel experiments are designed to verify our argument.

Our basic assumptions of this paper are as follows. Let the dust exposed on a bare soil surface be the available dust for aerodynamic entrainment. Then, the thoroughly disturbed soil possesses the maximum amount of available dust. As dust emission proceeds, supply limit for aerodynamic entrainment occurs when the available dust falls below a critical level. We define the replenishment of available dust as surface renewal. Then, saltation and creep enable surface renewal in several ways: (1) remove particles on the surface to expose the underlying dust; (2) spear into the soil to dislodge the dust initially not available; and (3) blast onto aggregates and break them to release new surface dust. Surface renewal does not directly cause dust emission but recover surface available dust, which is the main difference from normal saltation bombardment

mechanism. The total emitted dust is divided into two parts: one part is attributed to aerodynamic entrainment ($F_a$) and the other to sandblasting ($F_{b+c}$, including the contribution of saltation bombardment, $F_b$, and aggregates disintegration, $F_c$).

## 3. Wind Tunnel Experiment

We conducted the experiments in the wind tunnel of Lanzhou University. This open-return blow-down low-speed wind tunnel is 22 m long (only for work section) with a cross section of 1.3 m wide and 1.45 m high. The operational wind speed

can be adjusted in the range of 4-40 m·s$^{-1}$. The wind tunnel has excellent performance in simulating atmospheric boundary-layer flows for near-surface wind environment studies. The detailed information of the wind tunnel could be found in Zhang et al. (2014).

### 3.1 Experimental Setup

The setup for the experiments is as shown in Figure 1. Roughness elements are placed 6 m upstream the working section to

initiate a turbulent boundary layer. Their heights are adjusted to ensure a logarithmic wind profile (up to 20 cm above ground) in the downstream measurement area under all applied flow speeds. A test surface is located immediately downstream the roughness elements, which is 9 m long, 1 m wide and 5 cm deep and is paved with a soil. For measuring saltation, a sand trap is installed 8 m downstream from the frontal edge of the test surface. Two dust concentration probes are placed at 7 cm and 14 cm above the surface, each connected to a 1.109 Grimm aerosol spectrometer (Grimm Aerosol Technik GmbH & Co.

KG). A Pitot tube is anchored to an adjustable frame for measuring the profile of the flow speed at 10 sampling points at 10, 15, 20, 30, 50, 70, 100, 130, 160 and 200 mm above the surface.

A farmland soil collected from Minqin in Gansu Province of China (natural soil hereafter) and natural sand collected from the Tengger Desert (natural sand hereafter) are used for the preparation of the test surfaces. Three land surfaces are tested as shown in Figure 1. In Setting 1 (S1), the natural soil is used for the entire test bed to simulate a farmland surface, on which

supply limit may commonly occur. In Setting 2 (S2), the first 4 m of the test bed is paved with the natural sand ahead of 5 m natural soil, to examine how enhanced saltation affects dust emission with respect to S1. The S2 case corresponds to a desert-edge surface, on which saltation is significant to cause dust emission. In Setting 3 (S3), the natural soil is first sieved with a 20 mesh (841 μm) sieve (sieved soil hereafter) and then paved to simulate a loess surface which has sufficient dust content and low restriction for saltation.

### 3.2 Instruments and Measurements

By regression of the Prandtl–von Kármán equation

$$u(z) = \frac{u_*}{\kappa} ln\left(\frac{z}{z_0}\right) \tag{9}$$

to the Pitot-tube measurements, the friction velocity, $u_*$, and surface roughness, $z_0$, are estimated. In Equation (9), $z$ is height, $u(z)$ is the mean flow velocity (over three minutes in our experiments) at height $z$ and $\kappa=0.4$ is the von Kármán constant.

150 The particle size distributions of the natural soil, natural sand and sieved soil were analysed by using a Microtrac S3500 Laser Diffractometer (Microtrac, Montgomeryville, PA, USA) and approximated with an overlay of multiple log-normal distributions

$$d \times p(d) = \sum_{j=1}^{N} \frac{W_j}{\sqrt{2\pi}\sigma_j} exp[-\frac{(lnd-lnD_j)^2}{2\sigma_j^2}] \tag{10}$$

where $N$ is the number of distribution modes ($N \leq 4$), $W_j$ is the weight of the $j$th model of the particle size distribution, $D_j$ and
155 $\sigma_j$ are the parameters in the $j$th distribution. The particle size distribution of minimally disturbed soil $p_m(d)$ and fully disturbed soil $p_f(d)$ are measured similarly to Shao et al. (2011). The soil sample is dispersed in water and the resulting particle size distribution taken as $p_m(d)$. The soil is firstly ground in a mortar and then be dispersed in 2% sodium hexametaphosphate to prepare the measurement for $p_f(d)$. Although ultra-sonication is an effective method to break solid particles, the effect of chipping and attrition during particle collision does not occur during sonication which may result in wearing down individual
160 particles and changing the size distribution. Therefore, the sonication step in Shao et al. (2011) is replaced with grinding in measuring $p_f(d)$.

The saltation flux is measured using a sand trap adapted from the WITSEG sampler designed by Dong et al. (2003). Facing the wind stream are 38 stacked collectors (2 cm × 2 cm opening), each of which collects sand to its chamber. The streamwise saltation flux, $Q$, is then determined by weighing the sand in the chambers after each run:

165 $$Q = \sum_{i=1}^{38} q_i \Delta h_i \tag{11}$$

$$q_i = \frac{m_i}{t_s A_i} \tag{12}$$

where $\Delta h_i$ is the vertical size of inlet for collector $i$ mounted at height $h_i$ above the surface, $q_i$ is the saltation flux at $h_i$, $m_i$ is the mass of sand collected at $h_i$, $t_s$ is the time duration of sand collection and $A_i$ is the inlet area of the collector.

Once emitted, dust is transported vertically by turbulent diffusion. Assuming steady state and horizontal homogeneity, the
170 vertical diffusive flux is equal to dust emission flux and can be evaluated by the gradient method which has been applied in previous wind-tunnel studies on dust emission (Gillette et al., 1974; Fairchild and Tillery, 1982; Borrmann and Jaenicke, 1987). Our environmental wind-tunnel is designed for simulating atmospheric boundary layer flows and its performance has been validated by testing the pressure gradient and the stability of wind profile along with streamline. We also tested the performance of this wind-tunnel in simulating well-mixed dust cloud with an 8 m fetch in a previous study on dust
175 deposition (Zhang, 2013). Thus, the condition of our laboratory satisfies the requirements of the gradient method. In our experiments, dust concentration, $C$, is measured at $z_1 = 7$ cm and $z_2 = 14$ cm above the surface, and thus dust emission rate can be calculated as

$$F = -K_p \frac{C(z_2) - C(z_1)}{z_2 - z_1} \qquad (13)$$

where $K_p$ is the turbulent diffusion coefficient for dust particles, which can be approximated as

$$K_p = K_m = u_* l \qquad (14)$$

with $l$ being the mixing length, taken here as $\kappa(z_1+z_2)/2$. Except for the requirement of experimental condition mentioned above, the dust particles should be small enough, then the gravitational settling can be ignored and Equation (14) is applicable. Normally, this method works for particle with diameter smaller than 20 μm (Gillette et al., 1972; Sow et al., 2009; Shao et al., 2011).

### 3.3 Procedures of Wind-tunnel Experiments

The wind-tunnel experiments are carried out according to the settings given in Table 1 and the following procedures:

1. Prepare soil and pave test bed as shown in Figure 1;
2. Set up instruments as shown in Figure 1;
3. Set fan to target flow speed; measure dust concentration and wind speed over 10 minutes; end run early if test bed is blown bare or sand chambers are filled;
4. Turn off fan; record time duration for saltator collection; weigh mass of collected saltators; save dust concentration data measured with aerosol spectrometer;
5. Restart fan set to the same target speed as Step 3, and measure wind profile;
6. Remove paved soil (soil must not be reused because emission has changed dust content). Start over from Step 1 for next run.

### 4. Results and Analysis

### 4.1 Particle Size Distribution of Source Materials and Wind Profiles

The particle size distributions of the natural soil, natural sand and sieved soil are shown in Figure 2. The dots represent the measured values, while the lines Equation (10) fitted to the measurements (see Table 2 for fitting parameters). For the natural sand, the fraction of particles in the size range of 10-200 μm increases due to grinding, while for the natural soil and sieved soil the increased fractions are relevant to the size ranges of 1-10 μm and 30-60 μm.

The natural soil contained many lumps (diameter in centimetre scale) that can be easily broken by external impact or abrasion. These lumps disperse in water and thus the similarity in $p_m(d)$ between the natural and sieved soils does not reflect the existence of the large lumps in the natural soil. However, the lumps may significantly influence dust emission by causing spatial shear stress variations and by sheltering the surface from erosion. It was also found that the soil lumps were easily

destroyed during the sieving process and the characterization of large soil lumps remains a problem to be better solved in future research.

The wind velocities measured in the height range of 10-160 mm (the data obtained at the topmost measurement point are erratic and therefore not included) are shown in Figure 3. The dots are averaged wind speeds over 3 minutes measured with
210 Pitot tubes and the lines are the regressions using Equation (9). As shown, the profiles of the horizontal wind velocity follow the logarithmic law and can be well fitted with the Prandtl–von Kármán equation very well. The values of the regression parameters are listed in Table 1.

## 4.2 Streamwise Saltation Flux

The measured streamwise saltation fluxes are shown in Figure 4. For all three surfaces, saltation flux increased with friction,
but the saltation flux of S2 (natural soil surface under sand bombardment) was significantly larger than that of S1 (natural soil surface) by more than an order of magnitude, due to the impact of saltating sand particles. No saltation was detected over S1 and S2 for $u_* < 0.34$ m·s$^{-1}$. But over S3, significant saltation was measured for $u_* > 0.23$ m·s$^{-1}$. For $u_* > 0.35$ m·s$^{-1}$, the saltation flux of S3 obviously exceeded that of S1, but is smaller than that of S2.

It is necessary to validate first the formulations of streamwise saltation flux which is closely related to most dust emission
models (e.g. LS99, S04). In case of saltation of uniform particles, saltation flux can be estimated using the Owen model (i.e. Equation 8), but $c_0$ and $u_{*t}$ are tuneable parameters to be determined by regression to the observations. The model-simulated results are shown in Figure 4 (Regression 1, dotted curves) together with the regression parameters $c_0$ and $u_{*t}$ and determination coefficient, $R^2$. As $c_0$ is related to the terminal velocity of the saltating particles, it is obviously big for S2 (corresponding to big sand particles). $u_{*t}$ is effected by the size of soil particles and surface roughness, and is therefore large
for S1 (because of high surface roughness) and for S2 (because of big size of sand particles).

The above fitting is straightforward and gives reasonable results except for the cases when the friction velocity is close to the threshold friction velocity. An alternative method is to calculate the saltation fluxes for different particle size bin by Equation (8) and then integrate over the size bins to obtain the total saltation flux

$$Q = \int_{d_1}^{d_2} Q(d_s) \, p(d_s) \delta d_s \tag{15}$$

The threshold friction velocity is evaluated by (Shao and Lu, 2000)

$$u_{*t}(d_s) = \sqrt{A_n(\sigma_\varphi g d_{s+} \frac{r}{\rho d_s})} \tag{16}$$

where $A_n$ and $r$ are the regression parameters. The threshold friction velocities calculated using Equation (16), together with the regression parameters $A_n$ and $r$, are shown in Figure 4. It is seen that the second method (Regression 2, solid curves) gives a more accurate estimate of $Q$ than the first (Regression 1). And the threshold friction velocity appears to be influenced
by not only particle size but also surface conditions, as the different values of $A_n$ and $r$ imply. As shown in the inserted graph of Figure 4, the surmised curves of threshold friction velocity are different from the conventional threshold curve with a

minimum around 100 μm (Fletcher, 1976a, b; Greeley and Iversen, 1985; Shao and Lu, 2000). That divergence may be caused by saltation bombardment which may change the rule of threshold for different particles. Additionally, surface obstacles (such as lumps in S1 and S2) may also affect surface particle threshold by absorbing momentum and generating turbulent eddies.

It should be noted that for S1 and S3, the simulated surface is 8 m long in addition to the 6 m roughness section and therefore the saltation of soil particles should have been saturated, but not for S2 for which the simulated sand surface is only 4 m in length (Shao and Raupach, 1992; Rasmussen et al., 2015). For the unsaturated sand saltation, the particle speeds would increase with increased $u_*$ (Ho et al., 2011; Kok, 2011), which may cause the bombardment efficiency to increase.

### 4.3. Vertical Dust Flux

Vertical dust fluxes can be calculated with Equations (13) and (14) using the measured dust concentrations at the levels of 7 cm and 14 cm. In this study, dust is defined as particles with diameter smaller than 15 μm to satisfy the requirement of the gradient method. It can be seen from Figure 5 that for S1, dust emission has an initial sharp increase followed by a rapid decline (Figure 5a). The same phenomenon has been reported in earlier studies and is considered to be characteristic of aerodynamic entrainment under limited supply of free dust (Shao, 1993; Loosmore and Hunt, 2000). After 3 minutes, the vertical dust flux tends to be stable. Therefore, we calculated the average dust flux over the interval of 3 to 10 minute (dashed lines in Figure 5) for all cases and plotted the results in Figure 6 (triangles). For comparison, the data of LH2000 and MP2008 are plotted as circles and squares respectively. As shown, our results are comparable with MP2008 but obviously greater than LH2000. Generally, dust vertical fluxes increase with friction velocity by following a power function. But the results for the three surfaces differ by several orders of magnitudes.

By considering that S1 resembled the unperturbed surface in MP2008, whereas S2 and S3 the renewed surface in MP2008, the S2 surface was indeed renewed by external sand bombardment and the S3 by the spontaneous saltation and creep of big particles. Thus, the dust emission of S2 was about one order of magnitude larger than that of S1 because the former experienced stronger saltation bombardment. The dust flux of S3 was another order of magnitude larger than that of S1 because of the higher dust content at the surface.

In our experiments, paving the test bed caused mechanical disturbances to the soil. Thus, at the beginning of the run, the amount of free dust available for aerodynamic entrainment should be close to the maximum for the given soil. As dust emission continued, the amount of available free dust thus was gradually depleted and eventually exhausted. That appears to be a reasonable explanation of the phenomenon that occurred in S1 in the first three minutes. After about three minutes, dust emission was mainly attributed to weak saltation bombardment (Figure 5a). We therefore separate the time series into the two sections of 0-3 min and 3-10 min. The vertical dust flux averaged over the 0-3 min section, $F_{0\text{-}3min}$, is the dust emission due to both aerodynamic entrainment and saltation bombardment with unlimited dust supply. The dust flux averaged over the 3-10 min section, $F_{3\text{-}10min}$ ($F_b$), is the dust emission due to saltation bombardment under limited dust supply (here, the

effect of aggregates disintegration is not discussed individually and the related contribution is involved in $F_b$). Based on the theory of dust emission described in Section 2, dust emission via aerodynamic entrainment depends on the amount of exposed surface dust, and saltation bombardment dust relates to the dust content of subsurface. For the case without surface renewal (S1), as result of dust emission, the exposed surface dust was exhausted and supply-limit occured. But the dust content of the subsurface should not have changed significantly during the measurement time of 10 minutes, due to the lack of motion of large surface particles which renew the surface. So it is reasonable to assume that there was no significant difference in dust emission via saltation bombardment during the measurement time, and the difference between the average vertical dust fluxes over the first 3 minutes ($F_{0\text{-}3min}$) and over the last 7 minutes ($F_{3\text{-}10min}$) is therefore considered as the dust emission caused by aerodynamic entrainment ($F_a$) under unlimited dust supply (Figure 6, pentagram dots).

In contrast, S2 and S3 did not show such a remarkable decrease of dust flux after the initial phase, probably due to the intensive saltation (see Figure 4) which timely replenished the dust supply. But in general, it is not possible to separate the contributions due to aerodynamic entrainment and saltation bombardment. We have noted that the comparable saltation flux over S1 did not lead to surface renewal but over S3. This shows that surface renewal is affected both by saltation intensity and surface properties (i.e. S1 is more resistant to be renewed).

The results show that, the dust flux due to aerodynamic entrainment in S1 under unlimited supply was far greater than that in LH2000 (Figure 5) and the maximum value ($F_{a|max}$, which is about three times the average flux) even exceeded the dust flux due to strong saltation bombardment in S2. This may be due to the uneven distribution of surface shear over the rough surface in S1. Thus, we conclude that flow conditions, surface particle motion, dust availability and surface roughness can jointly cause dust fluxes to differ by orders of magnitudes.

The measured dust fluxes averaged over the period of 3-10 minutes are then examined with regression analysis. Equation (3) is chosen as the regression equation and the regression curves are shown as solid lines in Figure 6. For S1, the natural soil with weak saltation bombardment had a dust flux proportional to $u_*^4$, in agreement with Gillette and Passi (1988). The introduction of saltation bombardment in S2 increased dust emission by one order of magnitude, with dust flux proportional to $u_*^6$. In S3, dust flux increased by two orders of magnitude compared to S1, with dust flux proportional to $u_*^7$. But under unlimited supply in S1, the dust flux was proportional to $u_*^{10}$, if the threshold friction velocity is set to the same value as in the case of S1 with the period of 3-10 minutes (i.e. $u_{*t} = 0.29$ m·s$^{-1}$). The regression analysis shows that with intensified surface renewal from S1 to S3, the relationship between dust flux and friction velocity increasingly resembled the aerodynamic entrainment under unlimited supply. An interpretation of this could be that strong saltation bombardment and creep enabled surface renewal, thereby removing supply limit and maintaining dust emission at a high level. From this point of view, dust emission can be considered to be driven by a combination of aerodynamic entrainment and saltation bombardment. In consideration of that saltation and creep are responsible for surface renewal which restores the availability of dust for emission, the contribution of aerodynamic entrainment should not be ignored and may be dominant under some conditions.

To test the above hypothesis, the total dust vertical flux is considered as the sum of two parts

$$F = F_a + F_{b+c} = c_1 \cdot u_*^{10} \left(1 - \frac{u_{*t}}{u_*}\right) + c_2 \cdot u_*^4 (1 - \frac{u_{*t}}{u_*}) \tag{17}$$

where $c_1$ relates to exposed dust content and $c_2$ to subsurface dust content and impact energy of saltators. The first term on the right hand side of Equation (17) is attributed to aerodynamic entrainment and the second to saltation bombardment and aggregates disintegration. We now use Equation (17) to predict the vertical dust fluxes over the different surfaces. The values of $u_{*t}$ are assumed to be the same as in Figure 6 and $c_1$ and $c_2$ are obtained by regression analysis. As shown in Figure 7, Equation (17) can well describe the experimental data. And based on the estimated values of $c_1$ and $c_2$, the ratio of $F_a/F$ can be readily estimated, as shown in Figure 7 (dashed lines). It is seen that, sometimes (e.g. high $u_*$ over S2 and S3) the contribution of aerodynamic entrainment can exceed saltation bombardment ($F_a/F > 0.5$) and be the dominate mechanism for dust emission. It appears that saltation not only causes dust emission, but also surface renewal which restores the availability of dust for the emission.

### 4.4. Bombardment Efficiency

Bombardment efficiency, $\eta = F/Q$, (Gillette, 1979; Marticorena and Bergametti, 1995; Shao, 2008; Macpherson et al., 2008) is a key parameter for the saltation bombardment process. Previous studies suggested that dust emission is mainly due to saltation bombardment and for a given surface $\eta$ appears to be a relatively stable constant (Marticorena and Bergametti, 1995; Houser and Nickling, 2001). Others found that $\eta$ increases with $u_*$ (Nickling et al., 1999; Kok et al., 2012) and this increase depends on surface conditions (Shao, 2001). However, measurements are so far insufficient to verify this theory. In MP2008 (Macpherson et al., 2008), as the surface conditions were very complex, the measured bombardment efficiency scattered over a range of 4 orders of magnitude and did not show a fixed relationship with friction velocity.

The bombardment efficiencies we measured are shown as dots in Figure 8. It is observed that around the threshold friction velocity for each setting, $\eta$ ranged between 2.0 and 3.0 × $10^{-4}$ $m^{-1}$, which is close to the result of MP2008. However, it behaved differently as $u_*$ increased. In S1, it decreased exponentially with $u_*$. But in S2, $\eta$ firstly decreased and then increased with increasing $u_*$. For the case of S3, it monotonically increased with $u_*$.

We now analyse the possible reasons for the behaviour of $\eta$. In S1, the decrease cannot be explained using the existing dust emission modes (Lu and Shao, 1999; Shao, 2001, 2004). It is likely that as saltation bombardment was weak in S1 and could only lift the dust in a thin soil layer. Once the dust in this thin layer was depleted, the surface became dust supply limited. In S2, with the increase of $u_*$, the large amount of saltators from the upstream may have buried the dust on the surface of the test bed and changed its properties, thus leading to the decline in bombardment efficiency similar to S1. As $u_*$ further increased, the sand particles would not settle on the test bed, but continue to strike the surface and expose more dust to air, and thus increasing the bombardment efficiency. It implies that the degree of surface renewal may significantly affect the bombardment efficiency. In S3, the available dust content is high and the bombardment efficiency is much higher than that

in S1 and S2. The sieved soil used in S3, free from the sheltering of the lumps, is very mobile. Thus, as wind speed increased, the sieved soil particles may undertook strong bombardment over the surface and enhanced surface renewal. This allows an unlimited dust supply to maintain the bombardment efficiency. But even this does not seem to explain the increase of $\eta$ with exponent of $u_*$ (blue line in Figure 8). While the decline of $\eta$ with $u_*$ in S1 and the preceding stage of S2 may be due to the inadequate replenishment of dust supply, the increase of $\eta$ with $u_*$ in S3 and the last stage of S2 must be due to the contribution of aerodynamic entrainment, this appears to be in line with the previous discussion of Figure 7. We also note that the increase rate of $\eta$ with $u_*$ in the last stage of S2 is slight higher than that of S3. That should be caused by the unsaturated sand saltation in which the velocity of saltating particle may increase with $u_*$ (Ho et al., 2011; Kok, 2011) and thus the bombardment efficiency increases.

In short, we conclude that the strong saltation bombardment enabled surface renewal and dust supply to maintain saltation bombardment efficiency; if the surface renewal is inadequate, then $\eta$ decreases with $u_*$; in contrast, the saltation and creep generate sufficient surface renewal and hence dust supply, then $\eta$ increases with $u_*$.

## 5. Conclusions

Three soil surfaces, representing farmland, desert-edge and loess, were tested in a wind-tunnel experiment to examine the dust emission mechanisms. It has been found that:

(1) Flow conditions, saltation bombardment, surface dust content and ground obstacles may all significantly affect dust emission, causing dust emission to change over orders of magnitude;

(2) Dust emission due to aerodynamic entrainment from the natural soil surface is proportional to $u_*^{10}$, if the supply of free dust is unlimited, as in the initial phase (typically the first 2-3 minutes) of the wind-tunnel runs. This shows that in general, aerodynamic entrainment can be an important (even a dominant) process for dust emission under certain circumstances;

(3) Supply limit appears to be the major reason to restrict dust emission. In nature, dust emission may be often supply limited and hence the contribution of aerodynamic entrainment is determined by the renewal of the surface which results in increased availability of free dust for emission;

(4) Surface renewal through saltation and creep of surface particles should be the major pathway to ease the supply limit for dust emission. Surface renewal is not only important to the availability of dust for aerodynamic entrainment, but also important to the efficiency of saltation bombardment, $\eta$. It is shown that $\eta$ depends on friction velocity, and the dependency differs for different surfaces reliant on the process of surface renewal.

Dust emission seems to be a process driven by fluid motion and restricted by dust supply. The saltation and creep of large particles can generate surface renewal and restore the dust supply. Thus, the contribution of aerodynamic entrainment cannot be overlooked and the processes of supply limitation and surface renewal must be given due attention. Our experiment has shown that aerodynamic entrainment is highly efficient when dust supply is sufficient. Since surface renewal often does not

fully liberate the potential of aerodynamic entrainment, dust emission in general can be seen as limited aerodynamic
entrainment, and the extent of restriction depends on the degree of surface renewal.

This study does not contradict the earlier perception that saltation plays a fundamentally important role in dust emission, because saltation not only generates bombardment emission and aggregates disintegration, but also provides power for creep and contributes directly or indirectly to surface renewal. What is new in this paper is that we have been able to demonstrate the importance of surface renewal to aerodynamic entrainment in dust emission process.

In addition to the surface renewal by saltation and creep, or dynamic surface renewal, other processes, such as dust deposition and weathering, also contribute to surface renewal. Further experimental observations and theoretical analysis are necessary to establish a general surface renewal model.

**Acknowledgments**

This work is supported by the State Key Program of National Natural Science Foundation of China (91325203), the National
Natural Science Foundation of China (41371034, 11602100), the Innovative Research Group of the National Natural Science Foundation of China (11421062), the Fundamental Research Funds for the Central Universities (lzujbky-2014-1) and the Hebei Province Department of Education Found (QN2014111).

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

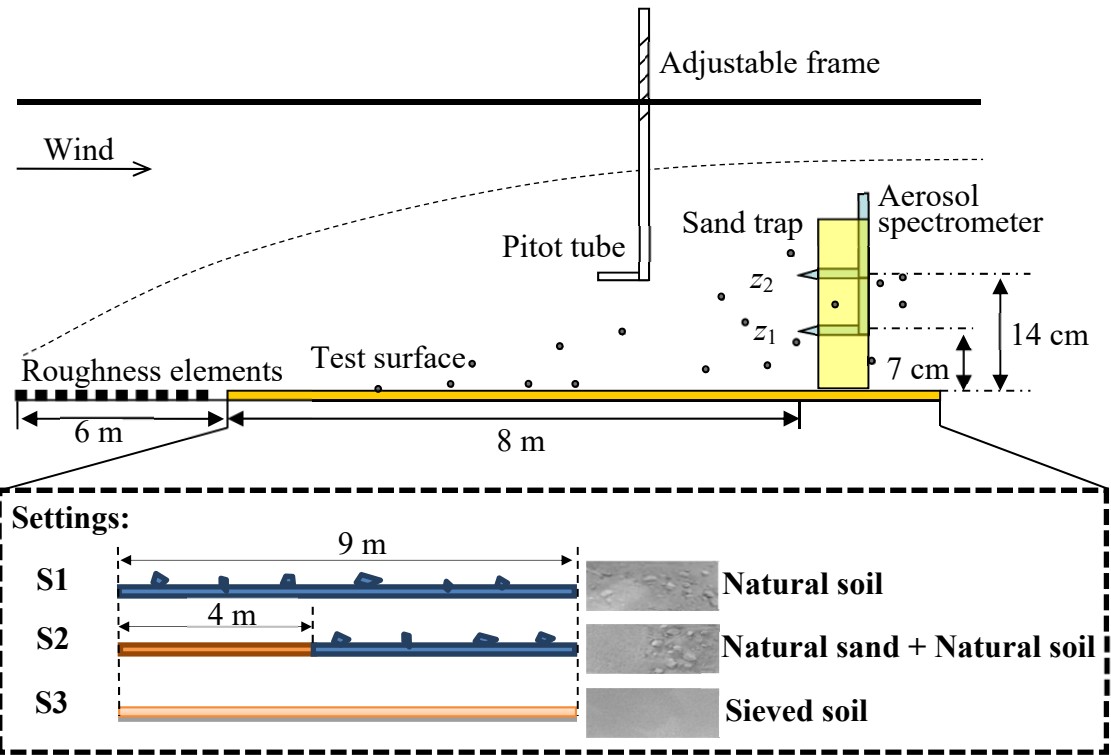

**Figure 1. Wind tunnel configuration and simulated soil surfaces. The test surface of 9 m long, 1 m wide and 5 cm deep is located immediately downstream the roughness elements. A position-adjustable Pitot tube is used to measure wind profile. A dust trap is installed 8 m downstream from the frontal edge of the test surface. Two GRIMM probes are fixed at 7 cm and 14 cm above the surface to measure dust concentration gradient.**

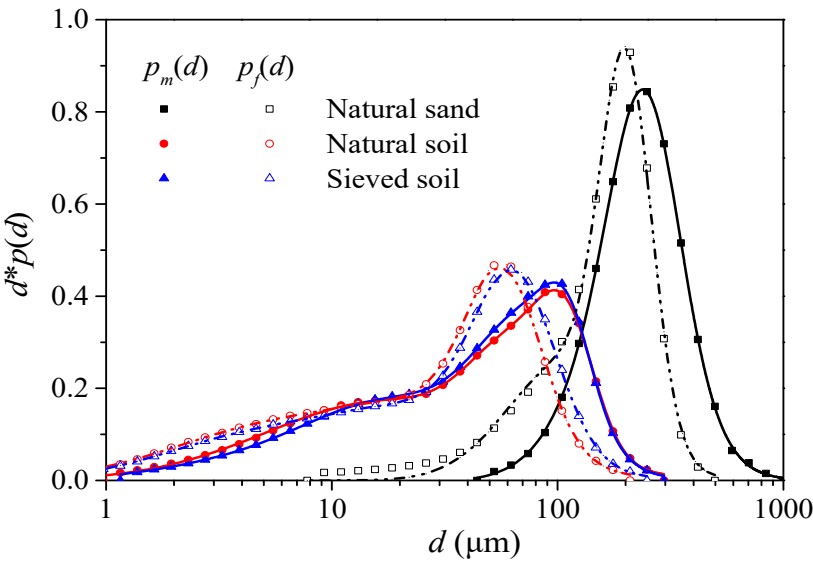

**Figure 2. Minimally- and fully-disturbed particle-size distributions of the source materials which are used to simulated the three surfaces illustrated in Figure 1, namely, the natural sand, natural soil and sieved soil. The dots represent the measured values, while the lines Equation (10) fitted to the measurements. The fitting parameters are shown in Table 2.**




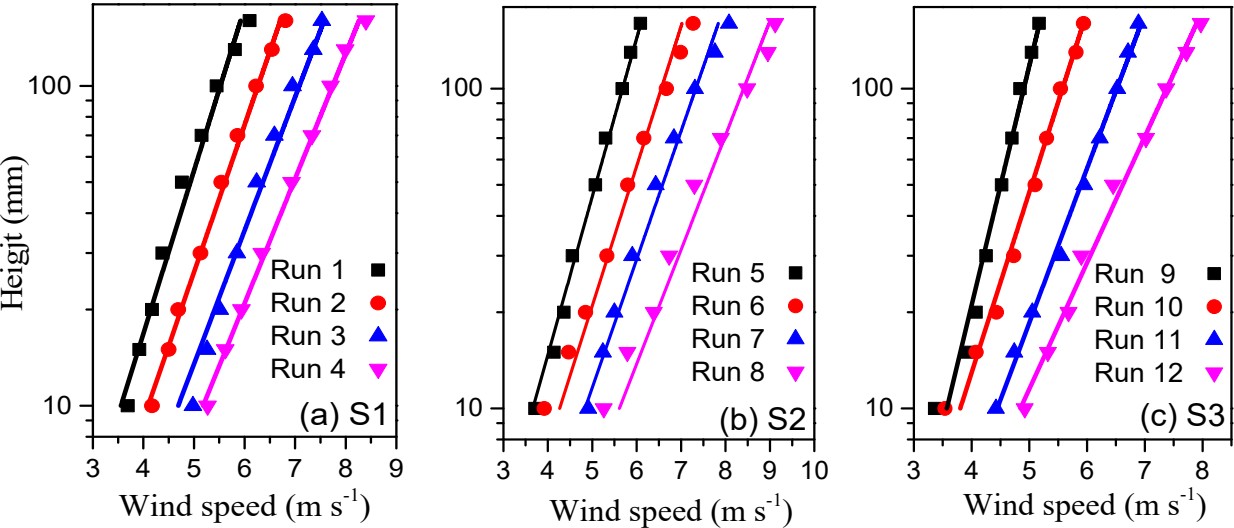

**Figure 3. Wind profiles over three different surfaces. The dots are experimental data and lines are regression curves from Equation (9). The regression parameters are listed in Table 1.**


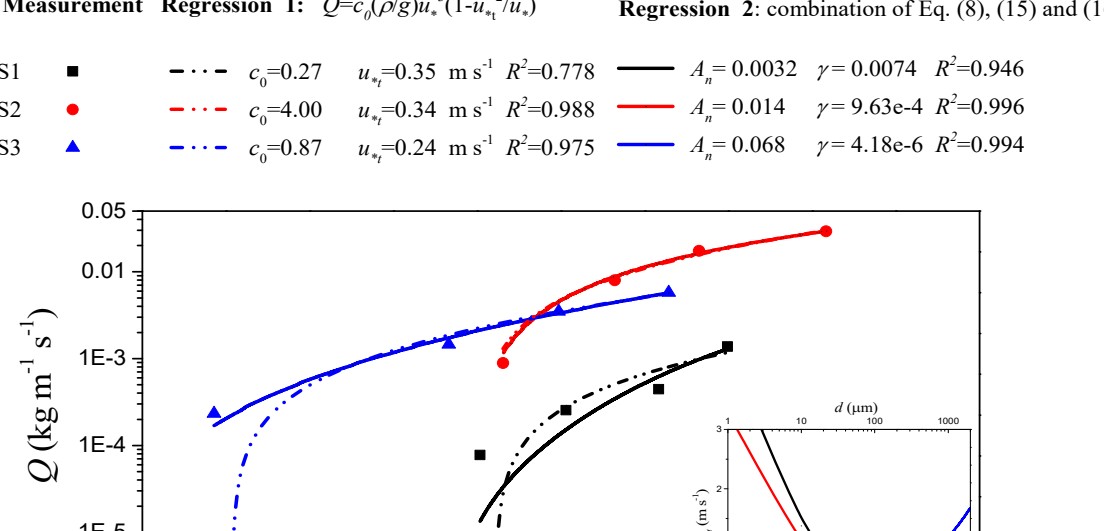

**Figure 4. Streamwise saltation flux over the three soil surfaces tested in the wind-tunnel experiment. The symbols are experimental data. The dot-dashed lines are regressions with Equation (8); $c_0$ and $u_{*t}$ are treated as regression parameters. The solid lines correspond to the combinations of Equation (8), (15) and (16). $A_n$ and $r$ are the regression parameters, which determine the friction velocities shown in the inserted graph.**

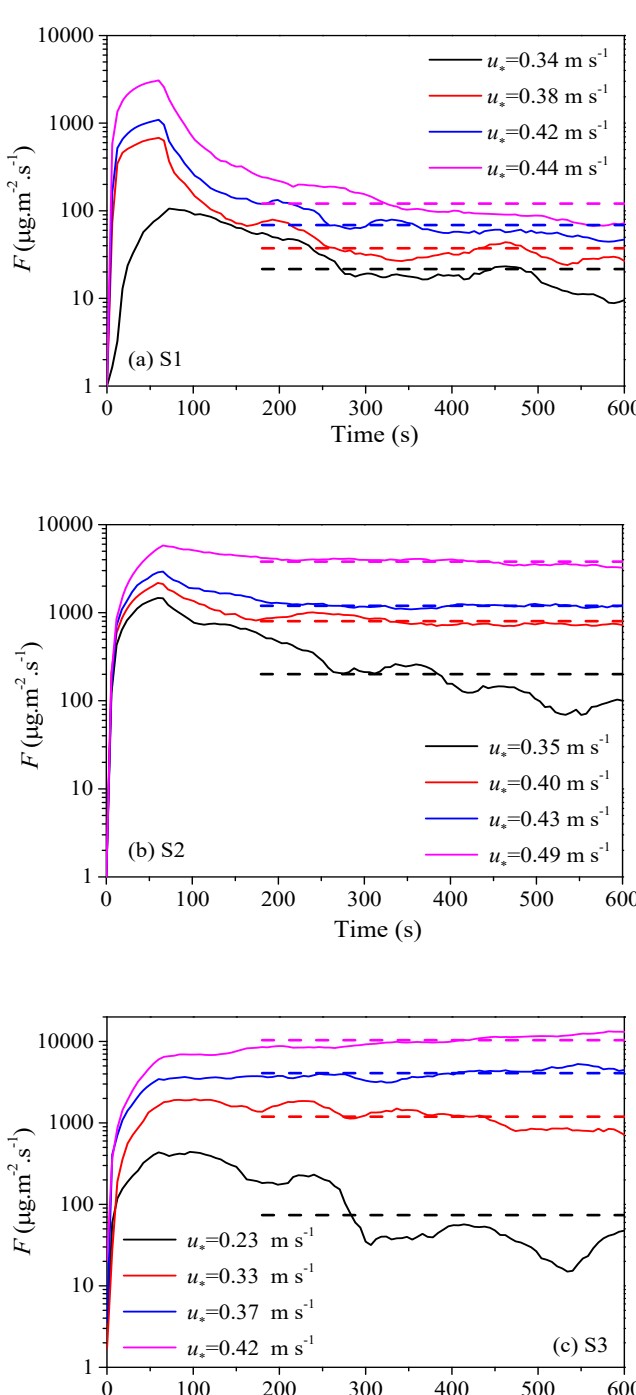

**Figure 5. Vertical dust flux series over the three surfaces tested in the wind-tunnel experiment. The dashed lines represent average**
**values form 3 to 10 minutes.**

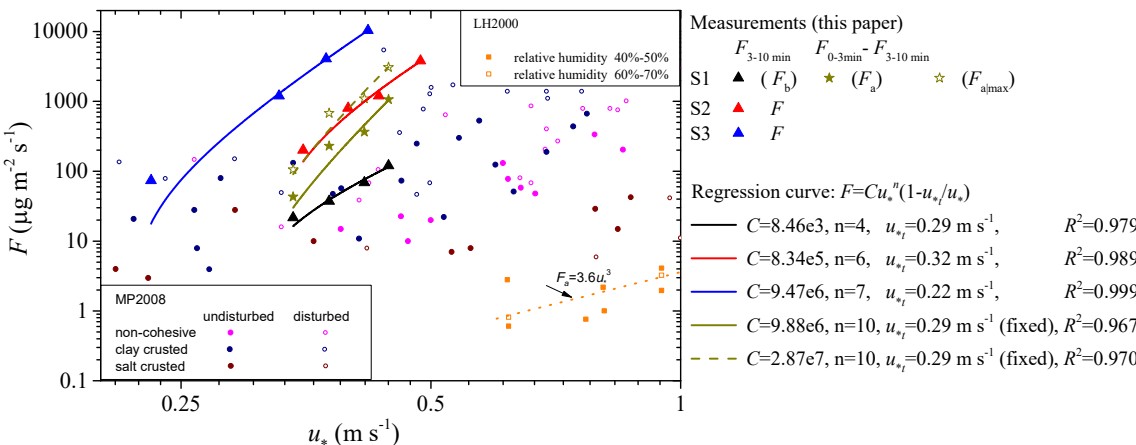

**Figure 6.** Measured vertical dust fluxes over the three different surfaces in the wind-tunnel experiment (triangles), together with the measurements of Loosmore & Hunt (2000) and Macpherson et al. (2008), respectively labeled as LH2000 and MP2008, as well as the various regression curves.

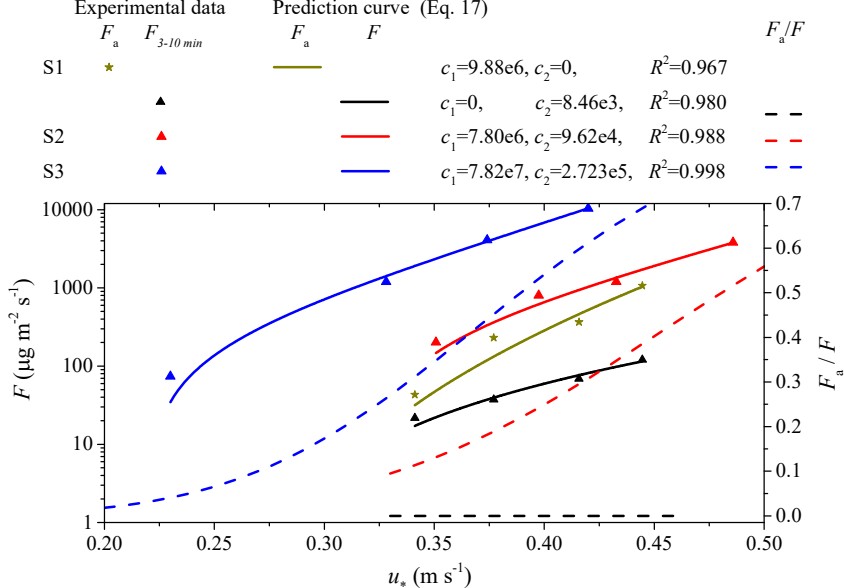


**Figure 7. Predictions of Equation (17) and the predicted contribution of aerodynamic entrainment $F_a$ is illustrated as the dashed lines with the right vertical coordinate. $u_{*t}$ is valued as the same to Figure 6. The solid lines and symbols are the same as in Figure 6.**


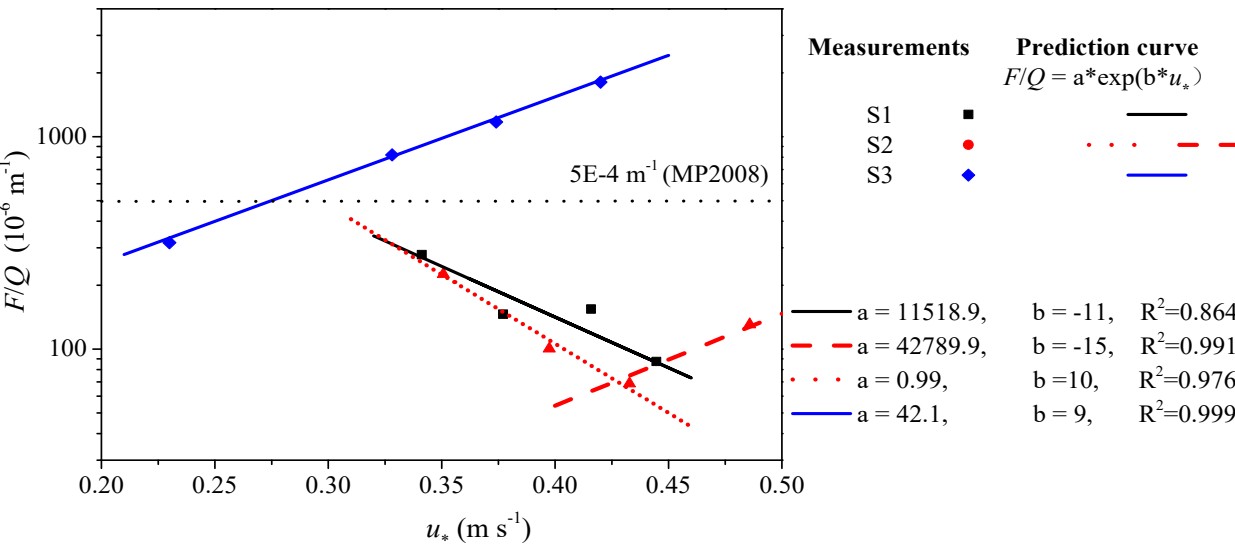

**Figure8. The ratio of dust emission to streamwise saltation flux. The symbols are experimental results and lines are prediction curves with the equation shown in the legend.**




**Table1. Runs for the dust-emission experiments and the regression parameters for wind profile over the three different surfaces. Four kinds of wind friction velocities are simulated for each surface. $R^2$ is determination coefficient of the regression.**

| Surface | runs | $u_*$ (m s$^{-1}$) | $z_0$(mm) | $R^2$ | Configuration |
|---------|------|--------|--------|------|---------------|
| S1 | 1 | 0.34 | 0.15 | 0.98 | Natural soil |
|    | 2 | 0.38 | 0.13 | 0.99 | |
|    | 3 | 0.42 | 0.11 | 0.98 | |
|    | 4 | 0.44 | 0.09 | 0.99 | |
| S2 | 5 | 0.35 | 0.15 | 0.99 | Natural soil +natural sand for bombardment |
|    | 6 | 0.40 | 0.14 | 0.97 | |
|    | 7 | 0.43 | 0.11 | 0.99 | |
|    | 8 | 0.49 | 0.10 | 0.95 | |
| S3 | 9 | 0.23 | 0.02 | 0.97 | Sieved soil |
|    | 10 | 0.33 | 0.10 | 0.98 | |
|    | 11 | 0.37 | 0.09 | 0.99 | |
|    | 12 | 0.42 | 0.09 | 0.99 | |





| Surface | runs | $u_*$ (m s$^{-1}$) | $z_0$(mm) | $R^2$ | Configuration |
|---------|------|--------|--------|------|---------------|

**Table 2. Log-normal distribution parameters for the three kinds of soils used in the experiments.** $d$ is particle diameter, $W_j$ is the weight of the $j$th distribution, $D_j$ and $\sigma_j$ are the parameters in the $j$th distribution, j ($\leq$4) refers to $j$th model.

| Material | | Mode 1 | | | Mode 2 | | | Mode 3 | | | Mode 4 | | |
|---|---|---|---|---|---|---|---|---|---|---|---|---|---|
| | | $W_1$ | $\ln(D_1)$ | $\sigma_1$ | $W_2$ | $\ln(D_2)$ | $\sigma_2$ | $W_3$ | $\ln(D_3)$ | $\sigma_3$ | $W_4$ | $\ln(D_4)$ | $\sigma_4$ |
| Sand | $p_m(d)$ | 0.471 | 5.51 | 0.34 | 0.529 | 5.34 | 0.54 | | | | | | |
| | $p_f(d)$ | 0.570 | 5.31 | 0.26 | 0.430 | 4.70 | 0.60 | | | | | | |
| Natural Soil | $p_m(d)$ | 0.196 | 4.70 | 0.29 | 0.229 | 4.42 | 0.43 | 0.575 | 2.88 | 1.23 | | | |
| | $p_f(d)$ | 0.357 | 4.06 | 0.37 | 0.314 | 3.44 | 0.86 | 0.329 | 1.73 | 1.06 | | | |
| Sieved Soil | $p_m(d)$ | 0.109 | 4.72 | 0.24 | 0.372 | 4.31 | 0.49 | 0.488 | 2.95 | 1.02 | 0.031 | 0.88 | 0.70 |
| | $p_f(d)$ | 0.408 | 4.17 | 0.41 | 0.364 | 3.29 | 0.92 | 0.228 | 1.49 | 0.94 | | | |
