# Peer review of "Surface Renewal as a Significant Mechanism for Dust Emission"

_Atmospheric Chemistry and Physics, 2016_

## Referee Comment (RC1) · Anonymous Referee #1 · 12 Aug 2016

General comments:

Zhang et al. 2016 present an analysis of wind tunnel experiments to examine the role of three dust emission mechanisms. While the authors try to address the role of surface renewal in dust emission, they also place much emphasis on the importance of aerodynamic entrainment compared to other emission mechanisms. I think this is the first study to address the surface renewal process based on wind tunnel experiments. I find that the paper is generally well written although some sections need to be restructured. The methods need improvements in several places. Often, I feel that the statements drawn by the authors lack sufficient evidence. More proper interpretations and in-depth discussions on the experimental data are needed in many places to support their conclusions. One big issue is that, although the wind tunnel experiments are well designed, the collected data sample is too small to tell real differences between

the regression fitting of various dust flux formulations. Plus, no statistics are given by the authors to judge the performance of regression analysis. I recommend publishing this paper after the following issues are addressed.

Detailed comments:

(P-page, L-line; note that the ACPD public version is used in this review)

My very first comment is that, please use continuous line numbering (instead of restarting numbering on every page) in your future manuscripts. This really helps the review process. Regardless whether the journal has such a requirement or not, using continuous line numbering is always a good practice.

Section 2 reads more like literature review, rather than a well-organized methods section. I encourage the authors to add a few sentences right after the section heading to explain how section 2 is organized before diving into the subsections. Another serious issue of section 2 is the use of symbols and abbreviations that are difficult to follow, because the authors give a review of so many dust schemes. Are they all necessary to be included in the paper? The authors need to make it clear why having an entire section for literature review of these specific dust flux methods is needed, and how they are going to connect with the wind tunnel experiments.

Equation 2: Why do the authors refer to the Gillette & Passi vertical flux parameterization, and then relate it to the Marticorena &Bergametti method of the F-Q relationship? Marticorena&Bergametti had their own parameterization for Q and F. That being said, equation 3 only applies for the Q parameterization in Marticorena&Bergametti, not necessarily the schemes from other studies.

Equation 4: Is this Fb or Fc? Later in eq. 8, you used Fc, but never defined Fc.

Equation 5-7: Should all the F in these equations be Fc? Also explain what F(di) and F(di, ds) are.

Equation 8 is questionable. My expression is that there are no distinct differentiations

between the three emission mechanisms in the model parameterizations. After all, they are mostly derived from wind tunnel experiment data, which most likely represent all three dust emission schemes. It is difficult to separate the different processes in field measurements or wind tunnel experiments. Even if Fa, Fb and Fc are specifically defined for the three processes, they formulations share the same parameters. However in fact, the validity of these formulations is only limited to certain conditions (e.g., wind speed, soil sizes), which are not discussed in the paper at all.

Equation 9: I think it is necessary to show a plot on regression analysis on calculating u* and z0 for all three experiments. Show the statistics from the regressions as well.

Equation 10: You never explained what the Pm(d) and Pf(d) are, and where they come from. I suppose that they come into play in Eq. 6. If so, define them after Eq. 6.

P6L3: How long does it take the fan to reach the target wind speed?

Section 3.1: Explicitly describe the purpose of the three experiments, for example, what dust emission mechanism(s) are each experiment corresponding to? What real-world conditions (e.g., supply limited in S1, supply limited but with renewal in S2, unlimited supply in S3?) do the experiments represent? I think having one or two statements like that can help readers easily understand the purpose of the experiment setup.

Equation 15 and Figure 3: Equations (15-19) should not be in the Results section. Move them to Section 2. I encourage the authors to rewrite Section 2 and logically introduce the dust schemes/equations (remove those not needed).

In Fig. 3: Why are there only 4 u*/Q values (same for other figures)? I donot think 4 runs for each surface type is sufficient to provide a meaningful data sample for regression analysis. Also, no statistics of regression (e.g., RMSE) are given. You can show them in the Fig.3. In P7L25, it is hard to "see" performance difference of two regression methods because of lack of statistical metrics. And the statistics should make more sense if a larger data sample is collected. If the wind tunnel can be configured to

reach any target wind speeds, it should not be difficult to make more measurements at variable wind speed conditions in order to collect a sufficiently large data sample. I think this is a big weakness of the paper.

P8L15: The way the aerodynamic entrainment is calculated (F0-3min minus F3-10min) is not convincing. Please explain why there is no significant difference between the saltation flux between 0-3 min (unlimited supply condition) and 3-10 min (supply limited condition)?

P9L1-7: This part of discussions is questionable. The authors state that "with intensified surface renewal from S1 to S3, the relationship between dust flux and friction velocity increasingly resembled the aerodynamic entrainment under unlimited supply." The authors show that the vertical dust flux is proportional to $u*^{10}$ in S1 strong saltation condition, $u*^4$ in S1 week saltation condition, $u*^6$ in S2, and $u*^7$ in S3. These n values still substantially deviates from the n=3 in Eq. 1. That means the F-u* relationship does not fall in the aerodynamic entrainment regime. By the rule, S1 supply limit state (n=4) is most close to the aerodynamic entrainment regime.

The authors also states that "From this point of view, dust emission can be considered to be mainly driven by aerodynamic entrainment, whereas saltation and creep are responsible for surface renewal which restores the availability of dust for emission. In general, dust emission can be seen as the result of restricted aerodynamic entrainment." I agree that saltation and creep is responsible for surface renewal; but that does not lead to the conclusion that during that process, aerodynamic entrainment is the main mechanism for dust emission. Saltation and aggregates disintegration are contributing to emission while they replenish the surface at the same time. The conclusion by the authors is not supported by any quantitative analysis that can prove the dominant role of aerodynamic entrainment in dust emission. Also, explain what 'restricted aerodynamic entrainment' means.

P9L32, the authors' claim that "the last stage of S2 must be due to the contribution of

aerodynamic entrainment" is not convincing. I understand that at high u* (last state of S2), surface renewal provides more erodible materials which increases the dust vertical flux. However, it is not necessarily due to the mechanism of aerodynamic entrainment. I think the authors are trying to emphasize the role of aerodynamic entrainment, but their analysis is groundless.

Considering the above comments, the abstract and conclusion sections of this paper must be rewritten. The summary #2 in the Section 5 is groundless and misleading. The authors state that n = 10 in the case of aerodynamic entrainment, but Eq. 1 shows n = 3 in the aerodynamic regime (Eq.1 was used throughout the paper to separate the aerodynamic entrainment regime). The authors state that aerodynamic entrainment is even a dominant process under certain circumstances. Please elaborate on that. What specific circumstances are they? I think the authors made lots of efforts to relate their experiments to aerodynamic entrainment, but the focus of this paper is on surface renewal. Many issues around that are not addressed, such as the renewal rate, dependence on wind speed/soil texture/soil size distribution/vegetation, biases in current dust schemes due to lack of surface renewal, and possible ways to introduce to dust schemes.

Minor comments:

Section 2.1, explicitly state that F is the vertical dust flux.

P1L24: there is -> there are.

P2L14: uplifted->uplift.

P2L15: inconsequential->insignificant.

P3L3: in equation 3, $\eta c$ is the soil clay content in percentage101.

Equation 4: $\eta$ is already used in eq.3, use a different symbol.

P3L20: you already defined u*t above.

P5L21: if -> of.

P5L25: use dust vertical flux (not dust emission rate) to be consistent throughout the manuscript.

P9L9: $\eta$ is already used in other places. use a different symbol.

P9L20: limit->limited.

Comments on Figures and figure/table captions:

Add S1, S2, S3 labels on the Fig. 2.

Add regression equations in Fig. 3.

Describe the horizontal dash lines in Fig. 4 caption.

Change 'dust emission' to 'dust vertical flux' in the Fig. 6 caption.

Table 2: Explain the meaning of the symbols in the caption (i.e. the parameters in the log normal size distribution).

[Figure]

---

## Referee Comment (RC2) · Anonymous Referee #2 · 29 Aug 2016

This is an interesting paper that uses wind-tunnel experiments to put forth the hypothesis that the renewal of fine particles in a soil's top layer is critical to sustaining dust emissions. This is an appealing hypothesis and this process is currently missing from models. They test this hypothesis using a series of wind tunnel measurements, which seem well designed. This article thus has the potential to be an important contribution.

However, there are several major issues with the article. Paramount is a major deficiency in how the results and discussion are presented. Almost throughout the "Results and Analysis" and "Conclusions" sections, the authors present hypotheses, of which their (otherwise very interesting) data are merely suggestive, as facts. Words such as "show" and "demonstrate" are used abundantly. This is not appropriate considering the level of evidence the authors present, and the enormous complexity of dust emissions. I will give a few (of many) examples of this below. The authors need to completely

rewrite these sections. In particular, they should split up the "Results" and "Discussion" sections, to make it clear what are indisputable facts from their experiments, and what is their interpretation of these facts.

In addition, there are some major scientific issues:

- The saltation bombardment section has major issues, which I'll list below:

* "c0 reflects the fraction of effective saltators, namely, grains available for saltation at a given friction velocity". This is inconsistent with Owen (1964), and also with the paper's own Eq. 18, where c0 is linked to the terminal velocity.

* P. 7, lines 10-11: I'm not aware of any measurements supporting the idea that the number of available saltators depends on the (theoretical) thresholds for individual particles. Rather, when saltation is initiated, the splashing process can mobilize particles of a wide range of sizes (e.g., Rice et al., 1995). The authors should either provide experimental evidence for their viewpoint, or note the opposing view (even if they do not adopt it).

* P. 7, lines 13-14: Did the authors directly measure what particles constituted the saltators? If not, this is interpretation, yet as presented as fact.

* P. 7, lines 14-15: This similarly is interpretation presented as fact. The saltation flux depends on many closely coupled and complex processes. Linking a change in the flux to any one parameter (the fraction of effective saltators in this case) without directly measuring it is speculative. That's fine to do in the discussion section, but should be presented as such.

* For the fitting with equation (15), how was u*t obtained? Was it fit as well? And how was v_t calculated?

* The use of Eq. (16) – (19) is very interesting. However, the procedure here is very unclear to me, and might have some scientific flaws. My primary concern is that the parameters in the u*t relation seem to be fit to the measurements, such that Eqs. (16) –

(19) have, as far as I can tell, three tunable parameters (proportionality constant (c0?), r, and An). Since the data they fit to are only four data points, these fits are statistically not that meaningful (only 1 degree of freedom). Thus the conclusion that "the above method gives a more accurate estimate of Q than Equation (15)" needs to be put on a more solid statistical basis.

* Related to the above comment, please provide the fitted u*t(d) relationships for the three soils so that the reader can judge whether they are reasonable. This is necessary to judge whether the visually good agreement is due to a good description of the physics, or because of a sufficient number of tuning parameters. You could provide these fits in a supplement to the paper.

- Sections 4.3 and 4.4: A central argument of the authors here is that the dust supply for aerodynamic entrainment is maintained by the intense sand flux for S2 and S3, but not for S1, which has lower sand flux at a given u*. However, the authors should compare apples to apples here and thus compare data with similar sand fluxes, for instance u* = 0.37 m/s for S1 and u* = 0.23 m/s for S3. The S1 data point shows a large dust flux decrease during the first minutes, whereas the S3 data point does not. This is not explained by their hypothesis, and should be clarified.

- I found section 4.4 very difficult to follow. Please use paragraphs in this section and make sure that the text flows smoothly. More importantly, this section again uses many interpretations of the data and would benefit enormously from separation into a results (facts) section and a discussion (interpretations and hypotheses) section. As it is written, I cannot sufficiently judge the scientific merit of this section.

- Section 4.5 suffers from similar issues as the other sections, with many hypotheses presented as though they were measured experimentally (line 9-11 "Due to the neglect of the supply-limiting effect and of the variation of bombardment efficiency, all three models underestimated the dust flux at low friction velocity, but slightly overestimated at high friction velocity"; line 14-15 "With the increase of u*, the bombardment

efficiency decreases because of changed surface property due to intrusive sand particles." ; line 18-19 "S04 appears to perform somewhat better than the others due to improved treatment for saltation bombardment and aggregates disintegration."; line 21-22 "This shows that threshold friction velocity $u*_t$ represents different properties of the soil surface in the Owen model and the GP88 model.")

Other comments:

- Please make line numbers continuous in revised article to make the review easier.

- In the literature I'm familiar with, the term "supply limited" is generally used to refer to a lack of supply of saltators, not a lack of supply of fine soil particles. The authors should clarify this point.

- Line 31-32, p. 1: Why do differences in dust emission after disturbing a soil indicate the importance of aerodynamic entrainment? This should be clarified or removed.

- Sections 2.2 and 2.3: While the authors cannot be expected to compare their data against every single dust emission model, they should at least mention the other ones (e.g., Marticorena and Bergametti (1995); Alfaro and Gomes (2001); Kok et al. (2014)).

- Eq. (9): What is the averaging time for $u(z)$?

- Line 15, p.5: This statement on sonification requires justification. For instance, the impact of saltating particles can chip and break them, which does not occur during sonification. Therefore, whereas sonification disaggregates particles, won't grinding result in the wearing down of individual (disaggregated) particles, thereby changing the size distribution?

- Please add a brief discussion whether the use of the gradient method is reasonable for your experiment. Compared to field measurements, your fetch is very small (a few meters, compared to 100s or 1000s of meters in the field). You partially compensated for this by moving your dust sensors close to the ground, but can you expect dust to be well-mixed (and thus follow a logarithmic profile) at only a few meters of fetch? How

will this affect your results?

- Section 3.3: was the wind flow seeded with particles in your experiments? If not, do you expect your sand flux to be saturated? The results of Shao and Raupach (1992) suggest that you need more length than the 8 m of your set-up.

- P. 7: please define d1 and d2 in Eq. 16. Also, the last d should be d_s

- P. 7: Please provide the value of the particle-to-air density

- In general, how exactly is the fitting performed? What quantity is minimized? Given that the data spans several orders of magnitude, it makes most sense to me to minimize the squared distance in log space, not in linear space (as the authors seem to have done).

- P. 8, line 3: does this refer to radius or diameter? Does this mean that the reported dust fluxes are limited to D (or r) < 15 um? Please clarify.

- P. 8, line 10-15: There are a lot of hypotheses used here to interpret the data in terms of arising from either aerodynamic entrainment or saltation bombardment, and whether or not the dust supply was limited. These factors were not measured directly, so these interpretations should be presented conservatively, rather than as statements of facts.

- P. 10: The scaling of aerodynamic entrainment with u* to the 10th power seems a bit extreme. Can you put uncertainty bounds on this result? How does this compare against other literature measurements such as Shao et al. (1993) and Loosmore and Hunt (2000)? What could explain the differences? Also, since you did not actually measure just aerodynamic entrainment (saltation was always present, as far as I understand), this conclusion should be more conservative.

- P. 11: "Supply limit is the major reason to restrict dust emission." This statement illustrates the main problem with the paper in its present form. Your measurements do not show this because you did not directly measure the supply limitations. You are merely hypothesizing this based on other measurements. I think it's a reasonable

hypothesis, but needs to be presented as such, and not as a fact or hard conclusion. This problem is persistent throughout the entire paper.

---

## Author Comment (AC1) · 31 Aug 2016

Response: we much appreciate the positive and insightful comments from the anonymous referee #1. These comments have motivated us to examine and revise the manuscript. Some sections of the manuscript will be restructured, according to the suggestions of referee. The details of responses are shown as following.

**Detailed comments:**
(P-page, L-line; note that the ACPD public version is used in this review)
My very first comment is that, please use continuous line numbering (instead of restarting numbering on every page) in your future manuscripts. This really helps the review process. Regardless whether the journal has such a requirement or not, using continuous line numbering is always a good practice.

Response: thanks for the reminding, the advice will be accepted.

Section 2 reads more like literature review, rather than a well-organized methods section. I encourage the authors to add a few sentences right after the section heading to explain how section 2 is organized before diving into the subsections. Another serious issue of section 2 is the use of symbols and abbreviations that are difficult to follow, because the authors give a review of so many dust schemes. Are they all necessary to be included in the paper? The authors need to make it clear why having an entire section for literature review of these specific dust flux methods is needed, and how they are going to connect with the wind tunnel experiments.

Response: thanks for the suggestions, section 2 will be restructured. Some sentences will be added before the section heading; the symbols and abbreviations will be checked and to be made clear.

Equation 2: Why do the authors refer to the Gillette & Passi vertical flux parameterization, and

then relate it to the Marticorena & Bergametti method of the F-Q relationship? Marticorena & Bergametti had their own parameterization for Q and F. That being said, equation 3 only applies for the Q parameterization in Marticorena & Bergametti, not necessarily the schemes from other studies.

Response: we listed typical achievement on vertical flux parameterization here. Marticorena & Bergametti considered F was a fraction of Q and the value of F/Q being imposed by the soil clay content. It was indeed that Marticorena & Bergametti had their own parameterization which was not employed in the manuscript. So we'd like to remove Eq.(3) to avoid misunderstanding.

Equation 4: Is this Fb or Fc? Later in eq. 8, you used Fc, but never defined Fc.

Response: Eq. (4) is the result of Fb and Eq. (7) including the contributions of Fb and Fc. We will make the Equations clear and define the variables before use.

Equation 5-7: Should all the F in these equations be Fc? Also explain what F(di) and F(di, ds) are.

Response: Eq. (7) including the contributions of Fb and Fc, we will make all of the variables clear.

Equation 8 is questionable. My expression is that there are no distinct differentiations between the three emission mechanisms in the model parameterizations. After all, they are mostly derived from wind tunnel experiment data, which most likely represent all three dust emission schemes. It is difficult to separate the different processes in field measurements or wind tunnel experiments. Even if Fa, Fb and Fc are specifically defined for the three processes, they formulations share the same parameters. However in fact, the validity of these formulations is only limited to certain conditions (e.g., wind speed, soil sizes), which are not discussed in the paper at all.

Response: that is true that dust emission mechanism is conceptually divided into three parts and it is hard to distinguish the contributions of these three sub-mechanisms from experimental data. Based on previous measurements, the vertical dust flux F is found to be proportional to $u_*^n$ with varying values of n. That may be caused by the different contributions of the sub-mechanisms under different conditions. But we still don't know the actual reasons in detail, which limits the knowledge of dust emission. In this paper, we design an experiment to separate the contributions of the three sub-mechanisms and should improve the understanding on dust emission. Anyway, it is acceptable that we need to add some necessary explanation and discussion in the paper.

Equation 9: I think it is necessary to show a plot on regression analysis on calculating u* and z0 for all three experiments. Show the statistics from the regressions as well.

Response: yes, we will add the results of wind profiles and the information of regression analysis.

Equation 10: You never explained what the Pm(d) and Pf(d) are, and where they come from. I suppose that they come into play in Eq. 6. If so, define them after Eq. 6.
Response: actually, Pm(d) and Pf(d) are defined in line 11-12 of page 6. The variable p(ds) in Eq.

6 is also related to Pm(d) and Pf(d). Some adjustments will be made.

P6L3: How long does it take the fan to reach the target wind speed?

Response: several seconds.

Section 3.1: Explicitly describe the purpose of the three experiments, for example, what dust emission mechanism(s) are each experiment corresponding to? What real-world conditions (e.g., supply limited in S1, supply limited but with renewal in S2, unlimited supply in S3?) do the experiments represent? I think having one or two statements like that can help readers easily understand the purpose of the experiment setup.
Response: accepted.

Equation 15 and Figure 3: Equations (15-19) should not be in the Results section. Move them to Section 2. I encourage the authors to rewrite Section 2 and logically introduce the dust schemes/equations (remove those not needed).
Response: accepted.

In Fig. 3: Why are there only 4 u*/Q values (same for other figures)? I do not think 4 runs for each surface type is sufficient to provide a meaningful data sample for regression analysis. Also, no statistics of regression (e.g., RMSE) are given. You can show them in the Fig.3. In P7L25, it is hard to "see" performance difference of two regression methods because of lack of statistical metrics. And the statistics should make more sense if a larger data sample is collected. If the wind tunnel can be configured to reach any target wind speeds, it should not be difficult to make more measurements at variable wind speed conditions in order to collect a sufficiently large data sample. I think this is a big weakness of the paper.

Response: our experiment is mainly limited by the amount of prepared soil material. To satisfy the requirement of experiment, the surface is made of soil material, with size of 9m length, 1m width and 5cm depth. Every surface is disposable and the soil material could not be used again, because of the changed dust content. That's why we only set 4 runs for each surface type. In the paper, we have 2 regression coefficients, and we thought the data of 4 runs were enough to run the regression analysis. But we still agree that the performance of regression analysis should be better, if more data sample is collected. The advice should be applied in future studies. Of course, we should add more statistics information of regression in this paper, such as the coefficient of determination $R^2$. Then, that will be easy to judge the performance difference of two regression methods

P8L15: The way the aerodynamic entrainment is calculated (F0-3min minus F3-10min) is not convincing. Please explain why there is no significant difference between the saltation flux between 0-3 min (unlimited supply condition) and 3-10 min (supply limited condition)?

Response: based on the definition of aerodynamic entrainment and saltation bombardment (as shown in the following picture), the emitted dust via aerodynamic entrainment depends on the amount of exposed surface dust, and saltation bombardment dust relates to the dust content of subsurface. For the case without surface renew (i.e. S1 in the paper), with the development of dust emission, the exposed surface dust is exhausted and supply-limit happens. But the content of subsurface should not change obviously during the total measurement time (10 mins), because of

few motion of big surface particle (no surface renew). So there should be acceptable to consider that there is no significant difference between the emission flux via saltation bombardment between 0-3 min and 3-10 min.

[Figure]

**Sketch map of (a) aerodynamic entrainment and (b) saltation bombardment**

P9L1-7: This part of discussions is questionable. The authors state that "with intensified surface renewal from S1 to S3, the relationship between dust flux and friction velocity increasingly resembled the aerodynamic entrainment under unlimited supply." The authors show that the vertical dust flux is proportional to $u*^{10}$ in S1 strong saltation condition, $u*^4$ in S1 week saltation condition, $u*^6$ in S2, and $u*^7$ in S3. These n values still substantially deviates from the n=3 in Eq. 1. That means the F-u* relationship does not fall in the aerodynamic entrainment regime. By the rule, S1 supply limit state (n=4) is most close to the aerodynamic entrainment regime.

Response: that may be a misunderstanding of the referee. The vertical dust flux, which is proportional to $u*^{10}$ in S1 (0-3 min, under unlimited supply), is actually caused by aerodynamic entrainment, but not saltation bombardment (the contribution of saltation bombardment has been subtracted). For the case of weak saltation condition (S1, 3-7 min), the vertical dust flux is proportional to $u*^4$ (only caused by saltation bombardment and aggregates disintegration, aerodynamic entrainment is exhausted because of supply-limit). For the case of strong saltation condition (S2, 3-7 min), the vertical dust flux is proportional to $u*^6$ (mainly caused by saltation bombardment, but includes the contribution of aerodynamic entrainment). And for the case of strong saltation and surface renew condition (S3, 3-7 min), the vertical dust flux is proportional to $u*^7$ (caused by saltation bombardment and aerodynamic entrainment). The value of n changes from 4 to 7, and is closed to 10 (aerodynamic entrainment under unlimited supply). So we state that "with intensified surface renewal from S1 to S3, the relationship between dust flux and friction velocity increasingly resembled the aerodynamic entrainment under unlimited supply." Eq. 1 (n=3) is not the reference of aerodynamic entrainment in our paper. And also we note that our result of aerodynamic entrainment dust is obviously bigger than the value of LH2000 (i.e. Eq.1, the comparison is shown in Fig. 5). That divergence may be caused by different experimental conditions, such as surface roughness and surface particle distribution. The exactly reason will be exposed in future study.

The authors also states that "From this point of view, dust emission can be considered to be mainly driven by aerodynamic entrainment, whereas saltation and creep are responsible for surface renewal which restores the availability of dust for emission. In general, dust emission can be seen as the result of restricted aerodynamic entrainment." I agree that saltation and creep is responsible for surface renewal; but that does not lead to the conclusion that during that process, aerodynamic entrainment is the main mechanism for dust emission. Saltation and aggregates disintegration are contributing to emission while they replenish the surface at the same time. The conclusion by the authors is not supported by any quantitative analysis that can prove the

dominant role of aerodynamic entrainment in dust emission. Also, explain what 'restricted aerodynamic entrainment' means.

Response: we will add some conceptual explanation for that. The regression curve of Fig. 5 (the modified version is shown in following) could be considered as the general formation for dust emission. The coefficient C relates to available dust content and the powder n relates to the mechanism of emission. Based on our measurements, n equals to 10 for aerodynamic entrainment and to 4 for saltation bombardment (also including the contribution of aggregates disintegration which is not identified in our work). Then the total dust flux could be expressed by

$$F = F_a + F_{b+c} = c_1 \cdot u_*^{10} \left(1 - \frac{u_{*\mathrm{t}}}{u_*}\right) + c_2 \cdot u_*^4 (1 - \frac{u_{*\mathrm{t}}}{u_*}) \qquad *$$

where $c_1$ relates to exposed dust content and $c_2$ to subsurface dust content and impact energy of saltators. The first part of the right of Eq. * is contributed by aerodynamic entrainment and the second part by saltation bombardment (and aggregates disintegration). We can use Eq. * to predict vertical dust flux over three different surfaces. The values of $u_{*\mathrm{t}}$ are the same as in Fig. 5 and $c_1$, $c_2$ could be obtained by regression analysis. As shown in Fig. *, the results of Eq. * agree with the experimental data very well. And based on the existing value of $c_1$, $c_2$, it's easy to give the ratio of $F_a/F$, shown as the dashed lines in Fig. *. The results show that, sometime (high $u_*$ over S2 and S3) the contribution of aerodynamic entrainment excess saltation bombardment ($F_a/F >50\%$). For that condition, aerodynamic entrainment becomes an important mechanism for dust emission. Saltation not only causes dust emission, but also is responsible for surface renewal which restores the availability of dust for emission (retrieve $c_1$ to a high level). For the first 3 mins of dust emission period over S1 (fully disturbed surface), aerodynamic entrainment is considered to be unlimited and $c_1$=9.88e6 is corresponding to 'fully aerodynamic entrainment'. For S2, the limitation of aerodynamic entrainment is relieved by intense saltation. But the value of $c_1$ (=7.80e6) is less than 9.88e6, which represents aerodynamic entrainment does not achieve the level of 'fully aerodynamic entrainment' and is considered as 'restricted aerodynamic entrainment'. Anyway, we will make some modifications for this part to ensure the conclusions are tenable.

[Figure]

**Figure 5: Measured dust emission fluxes over the three different surfaces in the wind-tunnel experiment (triangles), together with the measurements of Loosmore & Hunt (2000, LH2000) and Macpherson et al. (2008, MP2008), labeled as LH2000 and MP2008, as well as the various regression curves.**

[Figure]

**Figure \*: predictions of Eq. \* and the contribution of $F_a$ over different surfaces.**

P9L32, the authors' claim that "the last stage of S2 must be due to the contribution of aerodynamic entrainment" is not convincing. I understand that at high u\* (last state of S2), surface renewal provides more erodible materials which increases the dust vertical flux. However, it is not necessarily due to the mechanism of aerodynamic entrainment. I think the authors are trying to emphasize the role of aerodynamic entrainment, but their analysis is groundless.

Response: the results of Fig. \* shown above should support this statement.

Considering the above comments, the abstract and conclusion sections of this paper must be rewritten. The summary #2 in the Section 5 is groundless and misleading. The authors state that n = 10 in the case of aerodynamic entrainment, but Eq. 1 shows n = 3 in the aerodynamic regime (Eq.1 was used throughout the paper to separate the aerodynamic entrainment regime). The authors state that aerodynamic entrainment is even a dominant process under certain circumstances. Please elaborate on that. What specific circumstances are they? I think the authors made lots of efforts to relate their experiments to aerodynamic entrainment, but the focus of this paper is on surface renewal. Many issues around that are not addressed, such as the renewal rate, dependence on wind speed/soil texture/soil size distribution/vegetation, biases in current dust schemes due to lack of surface renewal, and possible ways to introduce to dust schemes.

Response: we didn't use Eq. 1 to separate the aerodynamic entrainment regime, but via $F_{0\text{-}3\ min}$ - $F_{3\text{-}10\ min}$ over S1. And as we stated before, the divergence between Eq. 1 and our results will be study in future work. The results of Fig. \* should be good to prove the statement 'aerodynamic entrainment is even a dominant process under certain circumstances' and the relevant expression will be rewritten.

We appreciate that the referee give us some good advices on surface renew research in future. But the main work of this paper is to point out the significant of surface renewal in dust emission mechanism. The detail study of surface renewal will be implemented in next.

**Minor comments:**

Section 2.1, explicitly state that F is the vertical dust flux.
Response: accepted.

P1L24: there is -> there are.
Response: accepted.

P2L14: uplifted->uplift.
Response: accepted.

P2L15: inconsequential->insignificant.
Response: accepted.

P3L3: in equation 3, $\eta_c$ is the soil clay content in percentage101.
Response: accepted.

Equation 4: $\eta$ is already used in eq.3, use a different symbol.
Response: accepted.

P3L20: you already defined u*t above.
Response: that will be removed.

P5L21: if -> of.
Response: accepted.

P5L25: use dust vertical flux (not dust emission rate) to be consistent throughout the manuscript.
Response: accepted.

P9L9: $\eta$ is already used in other places. use a different symbol.
Response: accepted.

P9L20: limit->limited.
Response: accepted.

Comments on Figures and figure/table captions:
Add S1, S2, S3 labels on the Fig. 2.
Response: accepted.

Add regression equations in Fig. 3.
Response: accepted.

Describe the horizontal dash lines in Fig. 4 caption.
Response: accepted.

Change 'dust emission' to 'dust vertical flux' in the Fig. 6 caption.
Response: accepted.

Table 2: Explain the meaning of the symbols in the caption (i.e. the parameters in the log normal size distribution).

Response: accepted.

---

## Author Comment (AC2) · 7 Sep 2016

**Response to anonymous referee #2's interactive comment on the manuscript "Surface Renewal as a Significant Mechanism for Dust Emission"**

This is an interesting paper that uses wind-tunnel experiments to put forth the hypothesis that the renewal of fine particles in a soil's top layer is critical to dust emissions. This is an appealing hypothesis and this process is currently missing from models. They test this hypothesis using a series of wind tunnel measurements, which seem well designed. This article thus has the potential to be an important contribution.

Response: we greatly appreciate the positive comments from referee 2#.

However, there are several major issues with the article. Paramount is a major deficiency in how the results and discussion are presented. Almost throughout the "Results and Analysis" and "Conclusions" sections, the authors present hypotheses, of which their (otherwise very interesting) data are merely suggestive, as facts. Words such as "show" and "demonstrate" are used abundantly. This is not appropriate considering the level of evidence the authors present, and the enormous complexity of dust emissions. I will give a few (of many) examples of this below. The authors need to completely rewrite these sections. In particular, they should split up the "Results" and "Discussion" sections, to make it clear what are indisputable facts from their experiments, and what is their interpretation of these facts.

Response: many thanks for the constructive comments which will be adopted in the revised version.

In addition, there are some major scientific issues:
- The saltation bombardment section has major issues, which I'll list below:
* "c0 reflects the fraction of effective saltators, namely, grains available for saltation at a given friction velocity". This is inconsistent with Owen (1964), and also with the paper's own Eq. 18, where c0 is linked to the terminal velocity.

Response: in Owen (1964) model, $c_0$ is well defined as a function of the ratio between the particle terminal velocity and the friction velocity for uniform particles, shown as Eq. 18. If this model is applied to mixed particles, we thought $c_0$ may be affected by other factors, such as particle size distribution. We try to give some explanations of $c_0$ for mixed-particle surface, and we agree with the referee that $c_0$ should follow its basic physical meaning to be consistent with Owen (1964). Some modification will be made for this part.

* P. 7, lines 10-11: I'm not aware of any measurements supporting the idea that the number of available saltators depends on the (theoretical) thresholds for individual particles. Rather, when saltation is initiated, the splashing process can mobilize particles of a wide range of sizes (e.g., Rice et al., 1995). The authors should either provide experimental evidence for their viewpoint, or note the opposing view (even if they do not adopt it).

Response: as stated above, we will rewrite this part to make the interpretation reasonable.

P. 7, lines 13-14: Did the authors directly measure what particles constituted the saltators? If not, this is interpretation, yet as presented as fact.

Response: we did not directly measure the component of saltators and the expression of this part will be changed.

P. 7, lines 14-15: This similarly is interpretation presented as fact. The saltation flux depends on many closely coupled and complex processes. Linking a change in the flux to any one parameter (the fraction of effective saltators in this case) without directly measuring it is speculative. That's fine to do in the discussion section, but should be presented as such.

Response: accepted, we will split up the part of "Results" and "Discussion" to make the expression clear and easy to follow.

* For the fitting with equation (15), how was u*t obtained? Was it fit as well? And how was v_t calculated?

Response: for the dashed line in Fig. 3, u*t and c0 were obtained from regression analysis with Eq. (15); for the solid line in Fig. 3, c0 was calculated by Eq. (18), and v_t is calculated by $v_t = 1.66(\sigma_\varphi g d_s)^{1/2}$ (Shao, 2008) which will be added in revised version.

The use of Eq. (16) – (19) is very interesting. However, the procedure here is very unclear to me, and might have some scientific flaws. My primary concern is that the parameters in the u*t relation seem to be fit to the measurements, such that Eqs. (16) –(19) have, as far as I can tell, three tunable parameters (proportionality constant (c0?), r, and An). Since the data they fit to are only four data points, these fits are statistically not that meaningful (only 1 degree of freedom). Thus the conclusion that "the above method gives a more accurate estimate of Q than Equation (15)" needs to be put on a more solid statistical basis.

Response: actually, here c0 is determined by Eq. (18) (v_t is calculated by $v_t = 1.66(\sigma_\varphi g d_s)^{1/2}$ as stated above) and there are only two tunable parameters (r and An) left. We will add the coefficient of determination $R^2$ to judge the performance of different regression methods.

* Related to the above comment, please provide the fitted u*t(d) relationships for the three soils so that the reader can judge whether they are reasonable. This is necessary to judge whether the visually good agreement is due to a good description of the physics, or because of a sufficient number of tuning parameters. You could provide these fits in a supplement to the paper.

Response: we can add the fitted u*t(d) relationships for the three soils. But actually, this work is not focus on testing existing scheme of saltation flux Q. We measured Q and searched a good formulation of Q to reduce the uncertainty in the validation of dust vertical flux F which is considered to be intimate with Q. that's why we didn't discuss too much about the reason of the visually good agreement in the paper.

- Sections 4.3 and 4.4: A central argument of the authors here is that the dust supply for aerodynamic entrainment is maintained by the intense sand flux for S2 and S3, but not for S1, which has lower sand flux at a given u*. However, the authors should compare apples to apples here and thus compare data with similar sand fluxes, for instance u* = 0.37 m/s for S1 and u* = 0.23 m/s for S3. The S1 data point shows a large dust flux decrease during the first minutes, whereas the S3 data point does not. This is not explained by their hypothesis, and should be clarified.

Response: generally speaking, dust supply for aerodynamic entrainment is maintained by the intense sand flux. But the criticality of sand flux which may cause surface renewal also depends on the property of surface. A certain sand flux may cause soft surface (e.g. S3) renewal, but may not efficient for hard surface (e.g. S1). That's the reason for the phenomenon that the referee mentioned above.

- I found section 4.4 very difficult to follow. Please use paragraphs in this section and make sure that the text flows smoothly. More importantly, this section again uses many interpretations of the data and would benefit enormously from separation into a results (facts) section and a discussion (interpretations and hypotheses) section. As it is written, I cannot sufficiently judge the scientific merit of this section.

Response: accepted, we will split up the part of "results" and "discussion" for this section.

- Section 4.5 suffers from similar issues as the other sections, with many hypotheses presented as though they were measured experimentally (line 9-11 "Due to the neglect of the supply-limiting effect and of the variation of bombardment efficiency, all three models underestimated the dust flux at low friction velocity, but slightly overestimated at high friction velocity"; line 14-15 "With the increase of u*, the bombardment efficiency decreases because of changed surface property due to intrusive sand particles." ; line 18-19 "S04 appears to perform somewhat better than the others due to improved treatment for saltation bombardment and aggregates disintegration."; line 21- 22 "This shows that threshold friction velocity u*t represents different properties of the soil surface in the Owen model and the GP88 model.")

Response: accepted, we will rewrite these sections carefully to make the expression reasonable and clear.

**Other comments:**
- Please make line numbers continuous in revised article to make the review easier.

Response: accepted.

- In the literature I'm familiar with, the term "supply limited" is generally used to refer to a lack of supply of saltators, not a lack of supply of fine soil particles. The authors should clarify this point.

Response: accepted.

- Line 31-32, p. 1: Why do differences in dust emission after disturbing a soil indicate the importance of aerodynamic entrainment? This should be clarified or removed.

Response: surface disturbance may change the content of exposed free dust which is intimate with aerodynamic entrainment. We will clarify this in revised version.

- Sections 2.2 and 2.3: While the authors cannot be expected to compare their data against every single dust emission model, they should at least mention the other ones (e.g., Marticorena and Bergametti (1995); Alfaro and Gomes (2001); Kok et al. (2014)).

Response: accepted.

- Eq. (9): What is the averaging time for u(z)?

Response: the wind speed is averaged over three minutes.

- Line 15, p.5: This statement on sonification requires justification. For instance, the impact of saltating particles can chip and break them, which does not occur during sonification. Therefore, whereas sonification disaggregates particles, won't grinding result in the wearing down of individual (disaggregated) particles, thereby changing the size distribution?

Response: accepted, we will add more appropriate explanation for the method selecting.

- Please add a brief discussion whether the use of the gradient method is reasonable for your experiment. Compared to field measurements, your fetch is very small (a few meters, compared to 100s or 1000s of meters in the field). You partially compensated for this by moving your dust sensors close to the ground, but can you expect dust to be well-mixed (and thus follow a logarithmic profile) at only a few meters of fetch? How will this affect your results?

Response: in fact, the gradient method has been employed in the other wind-tunnel studies on dust emission (Borrmann and Jaenicke, 1987; Fairchild and Tillery, 1982). Our environmental wind-tunnel laboratory is designed to simulate atmospheric boundary layer and has been validated. We did not test dust concentration profile in this experiment, because of the limitation of surface material. But the dust concentration profile has been tested in previous study of dust deposition (similar but with different transfer direction from emission), and the results shown the dust could be mixed well with the fetch of 8m in our wind-tunnel (Zhang, 2014). Anyway, it is accepted that we should add more discussion on rationality of gradient method.

- Section 3.3: was the wind flow seeded with particles in your experiments? If not, do you expect your sand flux to be saturated? The results of Shao and Raupach (1992) suggest that you need more length than the 8 m of your set-up.

Response: the wind flow was not seeded with particles. So we could not assure that the sand flux is saturated. That why we measured the saltation flux directly and strived to searched a good formulation of Q to reduce the uncertainty in the following validation of dust vertical flux F.

- P. 7: please define d1 and d2 in Eq. 16. Also, the last d should be d_s

Response: accepted.

- P. 7: Please provide the value of the particle-to-air density

Response: we will add it in the revised version.

- In general, how exactly is the fitting performed? What quantity is minimized? Given that the data spans several orders of magnitude, it makes most sense to me to minimize the squared distance in log space, not in linear space (as the authors seem to have done).

Response: we used 'Origin' (software) with the function of 'nonlinear curve fit' to implement data fitting. The iteration algorithm is set as 'Levenberg Marquardt'.

- P. 8, line 3: does this refer to radius or diameter? Does this mean that the reported dust fluxes are limited to D (or r) < 15 um? Please clarify.

Response: that refers to diameter and the reported dust fluxes are limited to D < 15 um. We will clarify them.

- P. 8, line 10-15: There are a lot of hypotheses used here to interpret the data in terms of arising from either aerodynamic entrainment or saltation bombardment, and whether or not the dust supply was limited. These factors were not measured directly, so these interpretations should be presented conservatively, rather than as statements of facts.

Response: accepted, we will check and rewrite this part carefully.

- P. 10: The scaling of aerodynamic entrainment with u* to the 10th power seems a bit extreme. Can you put uncertainty bounds on this result? How does this compare against other literature measurements such as Shao et al. (1993) and Loosmore and Hunt (2000)? What could explain the differences? Also, since you did not actually measure just aerodynamic entrainment (saltation was always present, as far as I understand), this conclusion should be more conservative.

Response: we will give more information of the regression analysis.
And actually we compared our results to Loosmore and Hunt (2000). The difference may be caused by different surface roughness.
Although saltation was always present, we subtract the contribution of saltation from total emission flux to obtain the quantity of aerodynamic entrainment.

- P. 11: "Supply limit is the major reason to restrict dust emission." This statement illustrates the main problem with the paper in its present form. Your measurements do not show this because you did not directly measure the supply limitations. You are merely hypothesizing this based on other measurements. I think it's a reasonable hypothesis, but needs to be presented as such, and not as a fact or hard conclusion. This problem is persistent throughout the entire paper.

Response: we will check the manuscript carefully and revise the relevant presentation.

---

## Author Response (AR1)

**Please note that the line numbers in this response is associated to the attached marked-up version.**

**Response to anonymous referee's #1**

**General comments:**

Zhang et al. 2016 present an analysis of wind tunnel experiments to examine the role of three dust emission mechanisms. While the authors try to address the role of surface renewal in dust emission, they also place much emphasis on the importance of aerodynamic entrainment compared to other emission mechanisms. I think this is the first study to address the surface renewal process based on wind tunnel experiments. I find that the paper is generally well written although some sections need to be restructured. The methods need improvements in several places. Often, I feel that the statements drawn by the authors lack sufficient evidence. More proper interpretations and in-depth discussions on the experimental data are needed in many places to support their conclusions. One big issue is that, although the wind tunnel experiments are well designed, the collected data sample is too small to tell real differences between the regression fitting of various dust flux formulations. Plus, no statistics are given by the authors to judge the performance of regression analysis. I recommend publishing this paper after the following issues are addressed.

Response: We much appreciate the positive and insightful comments from the anonymous referee #1. These comments have motivated us to examine and revise the manuscript. Some sections of the manuscript have been restructured, according to the suggestions of the referee. The details of responses are shown as following.

**Detailed comments:**

(P-page, L-line; note that the ACPD public version is used in this review)

My very first comment is that, please use continuous line numbering (instead of restarting numbering on every page) in your future manuscripts. This really helps the review process. Regardless whether the journal has such a requirement or not, using continuous line numbering is always a good practice.

Response: Thanks for the reminding, the advice has been accepted.

Section 2 reads more like literature review, rather than a well-organized methods section. I encourage the authors to add a few sentences right after the section heading to explain how section 2 is organized before diving into the subsections. Another serious issue of section 2 is the use of symbols and abbreviations that are difficult to follow, because the authors give a review of so many dust schemes. Are they all necessary to be included in the paper? The authors need to make it clear why having an entire section for literature review of these specific dust flux methods is needed, and how they are going to connect with the wind tunnel experiments.

Response: Thanks for the suggestions. Following the reviewer's comment, section 2 has been restructured (Line 49-143). The title of this section is changed to "Background of Dust Emission Mechanisms" (Line 49). Some sentences have been added before the section heading (Line 50-55); we deleted unnecessary equation and checked the symbols (Line 84-87, Line 93) and abbreviations to make the statement be clear.

Equation 2: Why do the authors refer to the Gillette & Passi vertical flux parameterization, and then relate it to the Marticorena & Bergametti method of the F-Q relationship? Marticorena & Bergametti had their own parameterization for Q and F. That being said, equation 3 only applies for the Q parameterization in Marticorena & Bergametti, not necessarily the schemes from other studies.

Response: We listed typical achievement on vertical flux parameterization here. Marticorena & Bergametti considered F was a fraction of Q and the value of F/Q being imposed by the soil clay content. Actually, Marticorena & Bergametti had their own parameterization, which was not employed in the manuscript. So we removed Eq. (3) to avoid misunderstand (Line 84-87).

Equation 4: Is this Fb or Fc? Later in eq. 8, you used Fc, but never defined Fc.

Response: Eq. (4) is the result of  $F_b$  and Eq. (7) including the contributions of  $F_b$  and  $F_c$ . We have made the equations clear and defined the variables in the revised manuscript (Line 50-53, Line 101).

Equation 5-7: Should all the F in these equations be Fc? Also explain what F(di) and F(di, ds) are.

Response: Thanks. Eq. (7) including the contributions of  $F_b$  and  $F_c$  (Line 101). We have made all of the variables clear.

Equation 8 is questionable. My expression is that there are no distinct differentiations between the three emission mechanisms in the model parameterizations. After all, they are mostly derived from wind tunnel experiment data, which most likely represent all three dust emission schemes. It is difficult to separate the different processes in field measurements or wind tunnel experiments. Even if Fa, Fb and Fc are specifically defined for the three processes, they formulations share the same parameters. However in fact, the validity of these formulations is only limited to certain conditions (e.g., wind speed, soil sizes), which are not discussed in the paper at all.

Response: Thanks. It was true that dust emission mechanism was only conceptually divided

into three parts for it was hard to distinguish the contribution of these three sub-mechanisms from experimental data. Based on previous measurements, the vertical dust flux F was found to be proportional to  $u_*^n$  with varying values of n, which was ascribed the different contributions of the sub-mechanisms under different conditions. But we still didn't know the actual reasons in detail, which limited the knowledge of dust emission. In this paper, we designed a serial of experiments to separate the contributions of the three sub-mechanisms, and thus to improve the understanding on dust emission. We agreed with the referee that the validity of the existing emission formulations was only limited to certain conditions (e.g., wind speed, soil sizes). Following the reviewer's comment, we have added some necessary explanations and discussions in lines 120-123.

But it appeared to be unnecessary to valid the formulations in this paper, which did not closely relate to the topic. Thus we deleted section 4.5 in the revised manuscript (Line 426-443).

Equation 9: I think it is necessary to show a plot on regression analysis on calculating u\* and z0 for all three experiments. Show the statistics from the regressions as well.

Response: Thanks. Following the reviewer's comment, we have added the results of wind profiles and the information of regression analysis (Line 236-240, 609-611 (Figure 3), 701-703 (Table 1)).

Equation 10: You never explained what the Pm(d) and Pf(d) are, and where they come from. I suppose that they come into play in Eq. 6. If so, define them after Eq. 6.

Response: Thanks for the careful reviewing. Following the reviewer's comment, some adjustments have been made in the revised manuscript and  $p(d_s)$ ,  $P_m(d)$  and  $P_f(d)$  are defined in Line 106-107

P6L3: How long does it take the fan to reach the target wind speed?

Response: It usually needs several seconds.

Section 3.1: Explicitly describe the purpose of the three experiments, for example, what dust emission mechanism(s) are each experiment corresponding to? What real-world conditions (e.g., supply limited in S1, supply limited but with renewal in S2, unlimited supply in S3?) do the experiments represent? I think having one or two statements like that can help readers easily understand the purpose of the experiment setup.

Response: Thanks. Following the reviewer's comment, we added the purpose of each experiment in the last paragraph of section 3.1 (Line 160-168).

Equation 15 and Figure 3: Equations (15-19) should not be in the Results section. Move them to Section 2. I encourage the authors to rewrite Section 2 and logically introduce the dust schemes/equations (remove those not needed).

Response: Thanks. Following the reviewer's comment, we have moved Equations (15-19) to section 2 and rewritten Section 2 to logically introduce the dust schemes/equations (Line 49-144).

In Fig. 3: Why are there only 4 u\*/Q values (same for other figures)? I do not think 4 runs for each surface type is sufficient to provide a meaningful data sample for regression analysis. Also, no statistics of regression (e.g., RMSE) are given. You can show them in the Fig.3. In P7L25, it is hard to "see" performance difference of two regression methods because of lack of statistical metrics. And the statistics should make more sense if a larger data sample is collected. If the wind tunnel can be configured to reach any target wind speeds, it should not be difficult to make more measurements at variable wind speed conditions in order to collect a sufficiently large data sample. I think this is a big weakness of the paper.

Response: Thanks. Our experiment was mainly limited by the amount of prepared soil material. To satisfy the requirement of experiment, the surface was made of soil material, with size of 9 m long, 1 m wide and 5 cm deep. Every surface was disposable and the soil material could not be used again, because of the changed dust content. That was why we only set 4 runs for each surface type. In the paper, we have 2 regression coefficients, and we thought the data of 4 runs were enough to run the regression analysis. But we still agreed that the performance of regression analysis should be better, if more data sample is collected. The advice should be applied in future studies. Following the reviewer's comment, we have added more statistics information of regression in the revised manuscript (for example, the coefficient of determination  $R^2$  in Figure 3, 4, 6, 7, 8) such that it would be easy to judge the performance difference of two regression methods

P8L15: The way the aerodynamic entrainment is calculated (F0-3min minus F3-10min) is not convincing. Please explain why there is no significant difference between the saltation flux between 0-3 min (unlimited supply condition) and 3-10 min (supply limited condition)?

Response: Thanks. Based on the definition of aerodynamic entrainment and saltation bombardment (as shown in the following picture), the emitted dust via aerodynamic entrainment depends on the amount of exposed surface dust, and saltation bombardment dust relates to the dust content of subsurface. For the case without surface renew (i.e. S1), as result of dust emission, the exposed surface dust is exhausted and supply-limit occurs. But the content of subsurface should not change significantly during the measurement time of 10 minutes, due to the lack of motion of large surface particle, which may renew surface. So it is reasonable to assume that there is no significant difference between the saltation bombardment emission flux during the period of 0-3 min and 3-10 min. Following the reviewer's comment, we added explanation for the data-processing method (Line 318-326)

P9L1-7: This part of discussions is questionable. The authors state that "with intensified surface renewal from S1 to S3, the relationship between dust flux and friction velocity increasingly resembled the aerodynamic entrainment under unlimited supply." The authors show that the vertical dust flux is proportional to  $u^{10}$  in S1 strong saltation condition,  $u^{4}$  in S1 week saltation condition,  $u^{6}$  in S2, and  $u^{77}$  in S3. These n values still substantially deviates from the n=3 in Eq. 1. That means the F-u\* relationship does not fall in the aerodynamic entrainment regime. By the rule, S1 supply limit state (n=4) is most close to the aerodynamic entrainment regime.

Response: Thanks. The vertical dust flux, which was proportional to u\*10 in S1 (0-3 min, under unlimited supply), was actually caused by aerodynamic entrainment for the contribution of saltation bombardment has been subtracted. For the case of weak saltation condition (S1, 3-10 min), the vertical dust flux was proportional to u\*4, which was only caused by saltation bombardment and aggregates disintegration for aerodynamic entrainment is exhausted because of supply-limit. For the case of strong saltation condition (S2, 3-10 min), the vertical dust flux was proportional to u\*6, which was mainly caused by saltation bombardment though a few contribution of aerodynamic entrainment was included. And for the case of strong saltation and surface renew condition (S3, 3-10 min), the vertical dust flux was proportional to u\*7, which was caused by both saltation bombardment and aerodynamic entrainment. The value of n changes from 4 to 7, and was closed to 10 for aerodynamic entrainment under unlimited supply. Based on above results we stated that "with intensified surface renewal from S1 to S3, the relationship between dust flux and friction velocity was more and more close to that of the aerodynamic entrainment under unlimited supply." Eq. 2 (n = 3) was not the reference of aerodynamic entrainment in our paper. And also we noted that our result of aerodynamic entrainment dust was obviously bigger than the value of LH2000 (i.e. Eq. 2, the comparison was shown in Fig. 6). That divergence may be caused by different experimental conditions, such as surface roughness and surface particle distribution. The exactly reason will be exposed in future study.

The authors also states that "From this point of view, dust emission can be considered to be mainly driven by aerodynamic entrainment, whereas saltation and creep are responsible for surface renewal which restores the availability of dust for emission. In general, dust emission can be seen as the result of restricted aerodynamic entrainment." I agree that saltation and creep is responsible for surface renewal; but that does not lead to the conclusion that during that process, aerodynamic entrainment is the main mechanism for dust emission. Saltation and aggregates disintegration are contributing to emission while they replenish the surface at the same time. The conclusion by the authors is not supported by any quantitative analysis that can prove the dominant role of aerodynamic entrainment in dust emission. Also, explain what 'restricted aerodynamic entrainment' means.

Response: Thanks. Following the reviewer's comment, we added some conceptual explanation for that (Line 345-360). The regression equation in Fig. 6 (as shown in following) could be considered as the general formation for dust emission. The coefficient C relates to available dust content and the powder n relates to the mechanism of emission. Based on our measurements, n equals to 10 for aerodynamic entrainment and to 4 for saltation bombardment (also including the contribution of aggregates disintegration which is not identified in our work). Then the total dust flux could be expressed by

$$F = F_a + F_{b+c} = c_1 \cdot u_*^{10} \left( 1 - \frac{u_{*t}}{u_*} \right) + c_2 \cdot u_*^4 \left( 1 - \frac{u_{*t}}{u_*} \right)$$
(17)

where  $c_1$  relates to exposed dust content and  $c_2$  to subsurface dust content and impact energy of saltators. The first term on the right hand side of Equation (17) is attributed to aerodynamic entrainment and the second to saltation bombardment and aggregates disintegration. We now use Equation (17) to predict the vertical dust fluxes over the different surfaces. The values of  $u_{*t}$  are assumed to be the same as in Figure 6 and  $c_1$  and $c_2$  are obtained by regression analysis. As shown in Figure 7, Equation (17) can well describe the experimental data. And based on the estimated values of  $c_1$  and $c_2$ , the ratio of  $F_a/F$  can be readily estimated, as shown in Figure 7 (dashed lines). It is seen that, sometimes (e.g. high  $u_*$  over S2 and S3) the contribution of aerodynamic entrainment can exceed saltation bombardment ( $F_a/F > 0.5$ ) and be the dominate mechanism for dust emission. It appears that saltation not only causes dust emission, but also surface renewal which restores the availability of dust for the emission.

Figure 6. Measured vertical dust fluxes over the three different surfaces in the wind-tunnel experiment (triangles), together with the measurements of Loosmore & Hunt (2000, LH2000) and Macpherson et al. (2008, MP2008), labeled as LH2000 and MP2008, as well as the various regression curves.

---

## Author Response (AR2)

**Response to anonymous referee's #1**

**Report #1**

Submitted on 21 Oct 2016
Anonymous Referee #1

**Anonymous during peer-review: Yes** No
**Anonymous in acknowledgements of published article: Yes** No

**Recommendation to the Editor**

| | |
|---|---|
| **1) Scientific Significance**
Does the manuscript represent a substantial contribution to scientific progress within the scope of this journal (substantial new concepts, ideas, methods, or data)? | **Excellent** Good Fair Poor |
| **2) Scientific Quality**
Are the scientific approach and applied methods valid? Are the results discussed in an appropriate and balanced way (consideration of related work, including appropriate references)? | **Excellent** Good Fair Poor |
| **3) Presentation Quality**
Are the scientific results and conclusions presented in a clear, concise, and well structured way (number and quality of figures/tables, appropriate use of English language)? | **Excellent** Good Fair Poor |

For final publication, the manuscript should be

**accepted subject to minor revisions**

Please note that this rating only refers to this version of the manuscript!

**Suggestions for revision or reasons for rejection (will be published if the paper is accepted for final publication)**

The authors made substantial efforts to address reviewers' comments and improve the presentation of their results. I therefore recommend publishing it after several issues are fixed below (minor revisions).

**Response: we sincerely thank the referee for the patience and perspicacious comments which are meaningful for the manuscript improvement and are also instructional for authors' future studies. Please find the detailed responses in the following sections.**

L150, describe here the averaging time for the measured u(z).

**Response:** thanks for the suggestion. The description of measured time for u(z) has been added in L150.

L179, What is the uncertainty of the gradient method in calculating F? Although the experimental setup prevents an accurate estimate of the uncertainty, the authors should provide some knowledge about the error sources of the method. Cite previous studies if necessary.

**Response:** thanks. the uncertainty of the gradient method is mainly caused by two aspects, including the environment conditions and the characteristic of dust particles. Since Equation (13) is essentially derived from dust conservation equation with the assumptions of steady state, horizontal homogeneity (results in constant vertical flux) and ignoring the effect of gravity, the gradient method is normally applied to measure fine dust (with diameter less than 20 μm) diffusion over large ground surface (several tens of meters) in field (Gillette et al., 1972; Sow et al., 2009). Alternatively, the gradient method also could be implemented in wind tunnel experiment (Gillette et al., 1974; Fairchild and Tillery, 1982; Borrmann and Jaenicke, 1987), if the turbulence boundary is completely developed and dusts are fully dispersed (to satisfy the requirement of steady state and horizontal homogeneity conditions) and the dust particle is small enough (then the particle can follow the air very well and the gravitational settling could be ignored). We had modified the descriptions of gradient method and added some knowledge about the error sources of this method in L170-185.

L197-198, fix the grammar error in this sentence.
**Response:** thanks, the error has been corrected.

Fig.1, for clarity, mark each colored line with the run # (1-12) in Table 1.

**Response:** thanks. But we guessed that the referee referred to Fig. 3, but not Fig. 1. And we have modified Fig. 3 according to the suggestion.

Table 1, why is the z0 for #9 (S3) so different from the rest?

**Response:** thanks. The value of $z_0$ was estimated form relevant wind speed profile. And we deduced that the various values of $z_0$ for S3 might be caused by the change of surface appearance. For the sieved soil S3 without lumps, the surface could be made very smooth in surface preparation process. And thus, $z_0$ was small for the case of low friction velocity (Run 9) with slight surface particle movement. But for the cases of high $u_*$ (Run 10-12), the surface particles movement became intense and the surface appearance changed. Some quasi ripples appeared and therefore $z_0$ changed to big value.

**Response to anonymous referee's #2**

**Report #2**

Submitted on 28 Oct 2016
Anonymous Referee #2

**Anonymous during peer-review: Yes** No

**Anonymous in acknowledgements of published article: Yes** No

**Recommendation to the Editor**

| | |
|---|---|
| **1) Scientific Significance**
Does the manuscript represent a substantial contribution to scientific progress within the scope of this journal (substantial new concepts, ideas, methods, or data)? | **Excellent** Good Fair Poor |
| **2) Scientific Quality**
Are the scientific approach and applied methods valid? Are the results discussed in an appropriate and balanced way (consideration of related work, including appropriate references)? | Excellent **Good** Fair Poor |
| **3) Presentation Quality**
Are the scientific results and conclusions presented in a clear, concise, and well structured way (number and quality of figures/tables, appropriate use of English language)? | Excellent **Good** Fair Poor |

For final publication, the manuscript should be

accepted subject to minor revisions

Please note that this rating only refers to this version of the manuscript!

**Suggestions for revision or reasons for rejection** (will be published if the paper is accepted for final publication)

The revisions to the paper have addressed most of my concerns. However, a few issues remain that should be addressed prior to publication:

**Response: we greatly appreciate the kind patience and insightful comments from the referee. All suggestions are earnestly considered and the detailed responses are shown as follow.**

• The authors justify the use of the gradient method on line 170-177. However, what does it mean that "Our environmental wind-tunnel is designed for simulating atmospheric boundary layer flows and its performance has been validated." How has it been validated? And it's still not clear that you have sufficient fetch that using the gradient method does not produce large errors. This needs more justification.

**Response:** thanks for the comment. The gradient method requires environmental condition to be steady and horizontal homogeneous. The method is thus normally applied in field of large flat ground. In wind tunnel, if the turbulence boundary lay is completely developed and the aeolian dusts are fully dispersed, the experimental condition could be considered to be satisfied with the above requirements and the gradient method therefore works. In our laboratory, we has tested the stability of averaged (over 3 mins) wind speed and dust concentration profiles (in previous work, Zhang 2013) over the measurement region (at the end of test surfaces) to validate the performance of our setup. The results shown that 8 m test surface added 6 m roughness elements (as shown in Fig. 1) is enough for turbulence development and dusts dispersion. We have added more justification for this part.

• The fits of the threshold friction velocity in Figure 4 seem unphysical. No experiment I'm familiar with has placed the minimum in the threshold friction velocity at either 15 um (for S3) or ~500 um (for S1). The authors should comment on whether these threshold are realistic, or if they are a consequence of maximizing the fit of an imperfect model to experimental data. If the former, what causes these threshold curves to be so different from the conventional threshold curve with a

minimum at ~100 um?

**Response:** thanks for this insightful comment. The threshold friction velocities in Fig. 4 were not directly measured but deduced from measured saltation flux and are indeed different from the conventional results. We try to give some explanations for the reason of the divergence. Typical studies on threshold friction velocity focused on fluid threshold over flat surface. But the threshold friction velocities in our paper essentially related to impaction threshold. The bombardment of saltation particles might change the rule of threshold for different particles and then modified the threshold curve (for the case of S3). Additionally, surface obstacles (such as lumps in S1 and S2) may also affect surface particle threshold velocity from mainly two aspects. One is to absorb momentum from flowing air and saltating particle as a shelter. The other one is to cause small eddied which may be positive for particle movement. We have added some discusses for the reason of the divergence in the revised version, although the accurate reasons are still not clear now.

• The authors need to acknowledge in the paper (they did so in the response) that the sand flux was probably not saturated. This means that particle speeds have not fully equilibrated to the wind flow, and thus that increasing u* will increase particle speeds, unlike what occurs in natural saltation. This will have affected the scaling of the measured dust emissions with wind speed. There are some recent works on this that should be discussed here (e.g., Shao and Raupach, JGR, 1992; Ho et al., PRL, 2011; Kok, ACP, 2011; Rasmussen et al., Geomorphology, 2015).

**Response:** thanks for the suggestion. Combining with the 6 m roughness elements before the 8 m test surface of S1 and S3, the saltation of soil particles should be saturated. But for S2, there were only 4 m sand surface. So the sand flux was probably not saturated and the particle speeds would increase with increased $u_*$, which may cause the bombardment efficiency to change. According to the referee's suggestion, we have added some discussions at the end of 4.2.

**List of changes** (the line numbers in the following list are associated to the attached marked-up version)

**Line 36**, 'subjected' was changed to 'subject'.

**Line 107**, 'physically based' was deleted.

**Line 109**, 'an indisputable fact is that' was deleted.

**Line 150**, '(over three minutes in our experiments)' was added after 'mean flow velocity'.

**Line 172**, 'Gillette et al., 1974;' was added before 'Fairchild and Tillery'.

**Line 174**, 'by testing the pressure gradient and the stability of wind profile along with streamline' was added after 'validated'.

**Line 182-185,** 'Except for the requirement of experimental condition mentioned above, the dust particles should be small enough, then the gravitational settling can be ignored and Equation (14) is applicable. Normally, this method works for particle with diameter smaller than 20 μm (Gillette et al., 1972; Sow et al., 2009; Shao et al., 2011).' was added at the end of Section 3.2.

**Line 201,** 'increased' was changed to 'increases'.

**Line 202**, '…in the 1-10 μm and 30-60 μm size range' was changed to '…the increased fractions are relevant to the size ranges of 1-10 μm and 30-60 μm'.

**Line 236-245,** some discusses were added at the end of Section 4.2 : 'As shown in the inserted graph of Figure 4, the surmised curves of threshold friction velocity are different from the conventional threshold curve with a minimum around 100 μm (Fletcher, 1976a, b; Greeley and Iversen, 1985; Shao and Lu, 2000). That divergence may be caused by saltation bombardment which may change the rule of threshold for different particles. Additionally, surface obstacles (such as lumps in S1 and S2) may also affect surface particle threshold by absorbing momentum and generating turbulent eddies.

It should be noted that for S1 and S3, the simulated surface is 8 m long in addition to the 6 m roughness section and therefore the saltation of soil particles should have been saturated, but not for S2 for which the simulated sand surface is only 4 m in length (Shao and Raupach, 1992; Rasmussen et al., 2015).  For the unsaturated sand saltation, the particle speeds would increase with increased $u_*$ (Ho et al., 2011; Kok, 2011), which may cause the bombardment efficiency to increase.'.

**Line 249-250**, ', such that the requirements for using the gradient method are satisfied (Shao et al., 2011)' was changed to 'to satisfy the requirement of the gradient method'.

**Line 340-343**, 'We also note that the increase rate of $\eta$ with $u_*$ in the last stage of S2 is slight higher than that of S3. That should be caused by the unsaturated sand saltation in which the velocity of saltating particle may increase with $u_*$ (Ho et al., 2011; Kok, 2011) and thus the bombardment efficiency increases.' was added at the end of this paragraph.

**Line 378,** ', 11602100' was added after '41371034'.

**Line 390**, 'Fletcher, B., The erosion of dust by an airflow, J. Phys. D Appl. Phys., 9(17), 913-924, 1976a. ' was added.

**Line 391**, 'Fletcher, B., The incipient motion of granular materials, J. Phys. D Appl. Phys., 9(17), 2471-2478, 1976b. ' was added.

**Line 401-402**, 'Ho T D, Valance A, Dupont P and A. Ould El Moctar, Scaling laws in aeolian sand transport, Phys. Rev. Lett., 106(9):265-270, 2011.' was added.

**Line 411-412**, 'Kok J., Does the size distribution of emitted dust aerosols depend on the wind speed at emission?, Atmos. Chem. Phys., 11, 10149–10156, doi: 10.5194/acp-11-10149-2011, 2011.' was added.

**Line 429-430**, 'Rasmussen K. R, Valance A., Merrison J., Laboratory studies of aeolian sediment transport processes on planetary surfaces. Geomorphology, 244:74-94, 2015.' was added.

**Line 435**, 'Shao Y, Raupach M R, The overshoot and equilibration of saltation. J. Geophys. Res., 97(D18):20559-20564, 1992.' was added.

**Line 451-452**, 'Sow M, Alfaro S. C., Rajot J. L. and Marticorena B., Size resolved dust emission fluxes measured in Niger during 3 dust storms of the AMMA experiment, Atmos. Chem. Phys., 9(12): 3881-3891, 2009.' was added.

**Line 515**, the information for each experiment was added in Fig. 3.

**The detailed modifications are shown in the marked-up version attached below.**

[revised manuscript text omitted]